# BCOS: A Method for Stochastic Approximation

## Abstract

We consider stochastic approximation with block-coordinate stepsizes and propose adaptive stepsize rules that aim to minimize the expected distance of the next iterate from an optimal point. These stepsize rules use online estimates of the second moment of the search direction along each block coordinate, and the popular Adam algorithm can be interpreted as using a particular heuristic for such estimation. By leveraging a simple conditional estimator, we derive variants of BCOS that obtain competitive performance but require fewer optimizer states and hyper-parameters. In addition, our convergence analysis relies on a simple aiming condition that assumes neither convexity nor smoothness, thus has broad applicability.

## 1 Introduction

We consider unconstrained stochastic optimization problems of the form

$$\underset{x \in \mathbf{R}^n}{\text{minimize}} \ F(x) := \mathbf{E}_\xi[f(x, \xi)], \tag{1}$$

where $x \in \mathbf{R}^n$ is the decision variable, $\xi$ is a random variable, and $f$ is the loss function. In the context of machine learning, $x$ represents the parameters of a prediction model, $\xi$ represents randomly sampled data, and $f(x, \xi)$ is the loss in making predictions about $\xi$ using the parameters $x$.

Suppose that for any pair $x$ and $\xi$, we can evaluate the gradient of $f$ with respect to $x$, denoted as $\nabla f(x, \xi)$. Starting with an initial point $x_0 \in \mathbf{R}^n$, the classical *stochastic approximation* method [38] generates a sequence $\{x_1, x_2, \ldots\}$ with the update rule

$$x_{t+1} = x_t - \alpha_t \nabla f(x_t, \xi_t), \tag{2}$$

where $\alpha_t$ is the *stepsize*, which is often called the *learning rate* in the machine learning literature. The convergence properties of this method are well studied in the stochastic approximation literature [e.g., 38, 3, 6, 44, 52]. Despite the rich literature on their convergence theory, stochastic approximation methods in practice often require heuristics and trial and error in choosing the stepsize sequence $\{\alpha_t\}$. Adaptive rules that can adjust stepsizes on the fly have been developed in both the optimization literature [e.g., 10, 25, 33, 40, 41, 42, 43] and by the machine learning community [e.g., 22, 32, 46, 47]. More recently, adaptive algorithms that use *coordinate-wise* stepsizes have become very popular following the seminal works of AdaGrad [14] and Adam [26]. In this paper, we present a framework for better understanding such methods and propose a family of new, effective methods.

### 1.1 Stochastic approximation with block-coordinate stepsizes

We focus on stochastic approximation with *block-coordinate stepsizes*, specifically of the form

$$x_{t+1} = x_t - s_t \odot d_t, \tag{3}$$

where $d_t \in \mathbf{R}^n$ is a stochastic search direction, $s_t \in \mathbf{R}^n$ is a vector of coordinate-wise stepsizes, and $\odot$ denotes element-wise product (Hadamard product) of two vectors. The two most common

choices for the search direction are: the *stochastic gradient*, i.e., $d_t = \nabla f(x_t, \xi_t)$, and its *exponential moving average (EMA)*. Let $g_t = \nabla f(x_t, \xi_t)$, the EMA of stochastic gradient can be expressed as

$$d_t = \beta d_{t-1} + (1 - \beta) g_t, \qquad (4)$$

where $\beta \in [0, 1)$ is a smoothing factor. This is often called the *stochastic momentum*.

The Adam algorithm [26] uses the direction in (4) and sets the coordinate-wise stepsizes as

$$s_{t,i} = \alpha_t / (\sqrt{v_{t,i}} + \epsilon), \qquad i = 1, \ldots, n, \qquad (5)$$

where $\alpha_t \in \mathbf{R}$ is a common stepsize *schedule* and each $v_{t,i}$ is the EMA of the squared coordinate gradient $g_{t,i}^2$, with a *different*, often *larger*, smoothing factor $\beta' \in (0, 1)$. More specifically,

$$v_{t,i} = \beta' v_{t-1,i} + (1 - \beta') g_{t,i}^2, \qquad i = 1, \ldots, n. \qquad (6)$$

Here $\epsilon > 0$ is a small constant to improve numerical stability when $v_{i,t}$ becomes very close to zero.

Adam [26] and its variant AdamW [31] have been very successful in training large-scale deep learning models. However, theoretical understanding of their convergence properties and empirical performance is still incomplete despite a lot of recent efforts [e.g., 37, 4, 1, 9, 56, 55, 28]. On the other hand, there have been many works that propose new variants or alternatives to Adam/AdamW, either starting from fundamental principles [e.g., 53, 17, 21, 29, 24] or based on empirical algorithm search [e.g., 5, 54] But all have limited success. Adam and especially AdamW are still the dominant algorithms for training large deep learning models, and their effectiveness remains a myth.

## 1.2 Contributions and outline

We propose a family of *block-coordinate optimistic stepsize* (BCOS) rules for stochastic approximation. BCOS provides a novel interpretation of Adam and AdamW and their convergence analysis as special cases of a general framework. Moreover, we derive variants of BCOS that obtain competitive performance but require fewer optimizer states and hyper-parameters. More specifically:

- In Section 2, we derive BCOS by minimizing the expected distance of the next iterate from an optimal point. While the optimal stepsizes cannot be computed exactly, we make optimistic simplifications and approximate the second moment of gradients with simple EMA estimators.

- In Section 3, we instantiate BCOS with specific search directions. In particular, we show that RMSprop [48] and Adam [26] can be interpreted as special cases of BCOS. By leveraging a simple conditional estimator, we derive new variants that require fewer optimizer states and hyper-parameters. Integrating with decoupled weight decay [31] gives the BCOSW variants.

- In Section 4, we present convergence analysis of BCOS(W) based on a simple aiming condition, which assumes neither convexity nor smoothness, thus has broad applicability. We obtain strong guarantees in terms of almost sure convergence, and characterize the effect of signal-to-noise ratio of the online estimators on the convergence behavior. Our results also apply to Adam(W).

- Finally, in Section 5, we present numerical experiments to compare BCOSW and AdamW on several Deep Learning tasks and demonstrate the effectiveness of the proposed methods.

## 1.3 Notations

Let $\mathcal{I}_1, \ldots, \mathcal{I}_m$ be a non-overlapping partition of the coordinate index set $\{1, \ldots, n\}$, each with cardinality $n_k = |\mathcal{I}_k|$. Correspondingly, we partition the vectors $x_t$, $s_t$ and $d_t$ into blocks $x_{t,k}$, $s_{t,k}$ and $d_{t,k}$ in $\mathbf{R}^{n_k}$ for $k = 1, \ldots, m$. We use a common stepsize $\gamma_{t,k} \in \mathbf{R}$ within each block, i.e., $s_{t,k} = \gamma_{t,k} \mathbf{1}_{n_k}$. As a result, the explicit block-coordinate update form of (3) can be written as

$$x_{t+1,k} = x_{t,k} - s_{t,k} \odot d_{t,k} = x_{t,k} - \gamma_{t,k} d_{t,k}, \qquad k = 1, \ldots, m.$$

Notice that $\gamma_{t,k}$ is always a scalar and $\gamma_t$ is a vector in $\mathbf{R}^m$ instead of $\mathbf{R}^n$ (unless $m = n$).

Throughout this paper, $\langle \cdot, \cdot \rangle$ denotes the standard inner product in $\mathbf{R}^n$ and $\| \cdot \|$ the induced Euclidean norm. The signum function is defined as $\text{sign}(\alpha) = 1$ if $\alpha > 0$, $-1$ if $\alpha < 0$ and $0$ if $\alpha = 0$.

## 2 Derivation of BCOS

We first derive the ideal optimal stepsizes for block-coordinate update, which is not computable in practice; then we make several simplifications and approximations to derive the practical ones.

## 2.1 Block-coordinate optimal stepsizes

We consider the change of distance to an optimal point $x_*$ after one iteration of the algorithm (3):

$$\|x_{t+1} - x_*\|^2 = \|x_t - s_t \odot d_t - x_*\|^2$$
$$= \|x_t - x_*\|^2 - 2\langle x_t - x_*, s_t \odot d_t \rangle + \|s_t \odot d_t\|^2.$$

Exploiting the block partitions of $x_t$, $s_t$ and $d_t$ and using $s_{t,k} = \gamma_{t,k}\mathbf{1}_{n_k}$, we obtain

$$\|x_{t+1} - x_*\|^2 = \|x_t - x_*\|^2 + \sum_{k=1}^{m} \left( -2\gamma_{t,k}\langle x_{t,k} - x_{*,k},\, d_{t,k} \rangle + \gamma_{t,k}^2\|d_{t,k}\|^2 \right).$$

Taking expectation conditioned on the realization of all random variables up to $x_t$, i.e.,

$$\mathbf{E}_t[\cdot] := \mathbf{E}[\cdot|x_0, d_0, x_1, d_1, \ldots, x_t], \tag{7}$$

we have

$$\mathbf{E}_t\left[\|x_{t+1} - x_*\|^2\right] = \|x_t - x_*\|^2 + \sum_{k=1}^{m}\left( -2\gamma_{t,k}\langle x_{t,k} - x_{*,k},\, \mathbf{E}_t[d_{t,k}]\rangle + \gamma_{t,k}^2\mathbf{E}_t\left[\|d_{t,k}\|^2\right]\right). \tag{8}$$

In order to minimize the expected distance from $x_{t+1}$ to $x_*$, we can minimize the right-hand side of (8) over the stepsizes $\{\gamma_{t,k}\}_{k=1}^{m}$. This results in the *optimal* stepsizes

$$\widehat{\gamma}_{t,k} = \frac{\langle x_{t,k} - x_{*,k},\, \mathbf{E}_t[d_{t,k}]\rangle}{\mathbf{E}_t[\|d_{t,k}\|^2]}, \qquad k = 1, \ldots, m. \tag{9}$$

Notice that these optimal stepsizes can be positive or negative, depending on the sign of the inner product in the numerator. Apparently, they are not computable in practice, because we do not have access of $x_*$ and cannot evaluate the expectations precisely. We address this issue in the next section.

## 2.2 Block-coordinate optimistic stepsizes

We need to make several simplifications and approximations to derive a practical stepsize rule. Our first step aims to avoid the direct reliance on $x_*$. To this end, we rewrite the numerator in (9) as

$$\langle x_{t,k} - x_{*,k},\, \mathbf{E}_t[d_{t,k}]\rangle = \|x_{t,k} - x_{*,k}\|\|\mathbf{E}_t[d_{t,k}]\|\cos\theta_{t,k},$$

where $\theta_{t,k}$ is the angle between the two vectors $x_{t,k} - x_{*,k}$ and $\mathbf{E}_t[d_{t,k}]$. We absorb the quantities related to $x_{*,k}$ into a tunable parameter $\alpha_{t,k} \approx \|x_{t,k} - x_{*,k}\|\cos\theta_{t,k}$, which gives the stepsizes

$$\widetilde{\gamma}_{t,k} = \frac{\alpha_{t,k}\|\mathbf{E}_t[d_{t,k}]\|}{\mathbf{E}_t[\|d_{t,k}\|^2]}, \qquad k = 1, \ldots, m. \tag{10}$$

We emphasize that any $\alpha_{t,k}$ we choose in practice may only be a (very rough) approximation of $\|x_{t,k} - x_{*,k}\|\cos\theta_{t,k}$. In particular, while the optimal stepsizes $\widehat{\gamma}_{t,k}$ can be positive or negative, in practice it is very hard to estimate the sign of the inner product $\langle x_{t,k} - x_{*,k},\, \mathbf{E}_t[d_{t,k}]\rangle$. Instead, we take the pragmatic approach of restricting $\alpha_{t,k} > 0$, effectively being *optimistic* that the expected search directions $-\mathbf{E}_t[d_{t,k}]$ always point towards $x_{*,k}$ for all $k = 1, \ldots, m$.

A further simplification is to use a common stepsize schedule $\alpha_t$ across all blocks. This is often a reasonable choice for deep learning, where the model parameters are initialized randomly coordinate-wise such that $\mathbf{E}[\|x_{0,k}\|]$ is constant for each coordinate $k$ [e.g., 13, 19]. This brings us to

$$\widetilde{\gamma}_{t,k} = \frac{\alpha_t\|\mathbf{E}_t[d_{t,k}]\|}{\mathbf{E}_t[\|d_{t,k}\|^2]}, \qquad k = 1, \ldots, m. \tag{11}$$

We note that with some abuse of notation, here $\alpha_t$ denotes a scalar, not a vector of $(\alpha_{t,1}, \ldots, \alpha_{t,k})$. This simplification reveals the connection between $\alpha_t$ and the distance $\|x_t - x_*\|$. Therefore, we expect $\alpha_t$ to decrease as $\|x_t - x_*\|$ gradually shrinks. A simple strategy is to use a monotonic stepsize schedule on $\alpha_t$, such as the popular cosine decay [30] or linear decay [8].

Next, we need to replace the conditional expectations $\mathbf{E}_t[d_{t,k}]$ and $\mathbf{E}_t[\|d_{t,k}\|^2]$ in (11) with computable approximations. We adopt the conventional approach of exponential moving average (EMA):

$$\begin{aligned} u_{t,k} &= \beta u_{t-1,k} + (1 - \beta)d_{t,k} \\ v_{t,k} &= \beta v_{t-1,k} + (1 - \beta)\|d_{t,k}\|^2 \end{aligned} \tag{12}$$

where $\beta \in [0, 1)$ is the smoothing factor. This leads to a set of practical stepsizes:

$$\gamma_{t,k} = \alpha_t\frac{\|u_{t,k}\|}{v_{t,k} + \epsilon}, \qquad k = 1, \ldots, m, \tag{13}$$

where we added a small constant $\epsilon > 0$ in the denominator to improve numerical stability.

| **Algorithm 1** BCOS-g | **Algorithm 2** BCOS-m |
|---|---|
| **input:** $x_0$, $\{\alpha_t\}_{t\geq 0}$, $\beta \in [0,1)$, $\epsilon > 0$ | **input:** $x_0$, $\{\alpha_t\}$, $\beta_1, \beta_2 \in [0,1)$, $\epsilon > 0$ |
| $v_{-1} = g_0^2$ | $m_{-1} = g_0$, $v_{-1} = g_0^2$ |
| **for** $t = 0, 1, 2, \ldots$ **do** | **for** $t = 0, 1, 2, \ldots$ **do** |
| $\quad g_t = \nabla f(x_t, \xi_t)$ | $\quad g_t = \nabla f(x_t, \xi_t)$ |
| $\quad v_t = \beta v_{t-1} + (1-\beta)g_t^2$ | $\quad m_t = \beta_1 m_{t-1} + (1-\beta_1)g_t$ |
| $\quad x_{t+1} = x_t - \alpha_t \frac{g_t}{\sqrt{v_t}+\epsilon}$ | $\quad v_t = \beta_2 v_{t-1} + (1-\beta_2)m_t^2$ |
| | $\quad x_{t+1} = x_t - \alpha_t \frac{m_t}{\sqrt{v_t}+\epsilon}$ |
| (same as RMSprop [49]) | |

## 2.3 Further simplification with one EMA estimator

The BCOS stepsizes in (13) are computed through the ratio of two online estimators $\|u_{t,k}\|$ and $v_{t,k}$, which are susceptible to large variations because the numerator and denominator may fluctuate in different directions. In this section, we derive a simplified stepsize rule that depends only on $v_{t,k}$.

First, recall the mean-variance decomposition of the conditional second moment,

$$\mathbf{E}_t[\|d_{t,k}\|^2] = \|\mathbf{E}_t[d_{t,k}]\|^2 + \mathbf{E}_t[\|d_{t,k} - \mathbf{E}_t[d_{t,k}]\|^2] = \|\mathbf{E}_t[d_{t,k}]\|^2 + \mathrm{Var}_t(d_{t,k}).$$

We interpret $\|\mathbf{E}_t[d_{t,k}]\|^2$ as the signal power and $\mathrm{Var}_t(d_{t,k})$ as the noise power, and define the *signal fraction* (SiF) as

$$\rho_{t,k} = \frac{\|\mathbf{E}_t[d_{t,k}]\|^2}{\mathbf{E}_t[\|d_{t,k}\|^2]} = \frac{\|\mathbf{E}_t[d_{t,k}]\|^2}{\|\mathbf{E}_t[d_{t,k}]\|^2 + \mathrm{Var}_t(d_{t,k})}. \tag{14}$$

Apparently we have $\rho_{t,k} \in [0,1]$. Using SiF, we can decompose the stepsizes in (10) as

$$\widetilde{\gamma}_{t,k} = \alpha_{t,k} \frac{\|\mathbf{E}_t[d_{t,k}]\|}{\mathbf{E}_t[\|d_{t,k}\|^2]} = \alpha_{t,k} \sqrt{\frac{\|\mathbf{E}_t[d_{t,k}]\|^2}{\mathbf{E}_t[\|d_{t,k}\|^2]}} \frac{1}{\sqrt{\mathbf{E}_t[\|d_{t,k}\|^2]}} = \frac{\alpha_{t,k}\sqrt{\rho_{t,k}}}{\sqrt{\mathbf{E}_t[\|d_{t,k}\|^2]}}. \tag{15}$$

Now we can merge $\sqrt{\rho_{t,k}} \in [0,1]$ into the tunable parameters $\alpha_{t,k}$ and let $\alpha'_{t,k} := \alpha_{t,k}\sqrt{\rho_{t,k}}$. Then, following the same arguments as in Section 2.2, we arrive at the following simplified stepsize rule:

$$\gamma_{t,k} = \alpha'_t \frac{1}{\sqrt{v_{t,k}}+\epsilon}, \qquad k = 1, \ldots, m, \tag{16}$$

where $\alpha'_t$ is a *scalar* stepsize schedule, and $v_{t,k}$ is given in (12). The similarity between Adam and BCOS in (16) is apparent, and we will explain their connection in detail in the next section.

## 3 Instantiations of BCOS

The derivation of BCOS in Section 2 is carried out with a general search direction $d_t$. In this section, we instantiate BCOS with two common choices of the search direction: the stochastic gradient and its EMA, also known as the *stochastic momentum*.

To simplify presentation, we focus on the case of *single coordinate blocks*, i.e., $m = n$ and $\mathcal{I}_k = \{k\}$ for $k = 1, \ldots, n$. Then we can express the EMA estimators for $\mathbf{E}_t[d_{t,k}^2]$ in a vector form:

$$v_t = \beta v_{t-1} + (1-\beta)d_t^2, \tag{17}$$

where $d_t^2$ denotes the element-wise squared vector $d_t \odot d_t$. We also have $s_t = \gamma_t \in \mathbf{R}^n$ and therefore

$$x_{t+1} = x_t - \gamma_t \odot d_t,$$

where the vector of coordinate-wise stepsizes, $\gamma_t$, can be expressed as

$$\gamma_t = \alpha_t \frac{1}{\sqrt{v_t}+\epsilon}. \tag{18}$$

Here $\sqrt{v_t}$ denotes element-wise square roots, $\sqrt{v_t} + \epsilon$ means element-wise addition of $\epsilon$, and the fraction represent element-wise division or reciprocal. Again, the stepsize schedule $\alpha_t$ is a scalar. We no longer distinguish between $\alpha_t$ and $\alpha'_t$ because they are both tunable hyper-parameters.

| **Algorithm 3** BCOS-c | **Algorithm 4** BCOSW-c |
|---|---|
| **input:** $x_0$, $\{\alpha_t\}_{t\geq 0}$, $\beta \in [0,1)$, $\epsilon > 0$ | **input:** $x_0$, $\{\alpha_t\}_{t\geq 0}$, $\beta \in [0,1)$, $\epsilon > 0$ |
| $m_{-1} = g_0$, $v_{-1} = g_0^2$ | $m_{-1} = g_0$, $v_{-1} = g_0^2$ |
| **for** $t = 0, 1, 2, \ldots$ **do** | **for** $t = 0, 1, 2, \ldots$ **do** |
| $\quad g_t = \nabla f(x_t, \xi_t)$ | $\quad g_t = \nabla f(x_t, \xi_t)$ |
| $\quad m_t = \beta m_{t-1} + (1-\beta) g_t$ | $\quad m_t = \beta m_{t-1} + (1-\beta) g_t$ |
| $\quad v_t = \big(1 - (1-\beta)^2\big) m_{t-1}^2 + (1-\beta)^2 g_t^2$ | $\quad v_t = \big(1 - (1-\beta)^2\big) m_{t-1}^2 + (1-\beta)^2 g_t^2$ |
| $\quad x_{t+1} = x_t - \alpha_t \frac{m_t}{\sqrt{v_t}+\epsilon}$ | $\quad x_{t+1} = (1 - \alpha_t \lambda) x_t - \alpha_t \frac{m_t}{\sqrt{v_t}+\epsilon}$ |

### 3.1  BCOS with EMA estimators

**BCOS-g**  Algorithm 1 is the instantiation of BCOS using $\nabla f(x_t, \xi_t)$ as the search direction. We call it BCOS-g to signify the use of gradient as search direction. The vector $v_t$ consists of coordinate-wise EMA estimators for $\mathbf{E}[g_{t,k}^2]$, and the notation $\frac{m_t}{\sqrt{v_t}+\epsilon}$ means element-wise division.

We immediately recognize that BCOS-g is exactly the RMSprop algorithm [49], which is one of the first effective algorithms to train deep learning models. Our BCOS framework gives a novel interpretation of RMSprop and its effectiveness. In the special case with $\beta = 0$ and $\epsilon = 0$, we have $v_t = g_t^2$, and both BCOS-g becomes the sign gradient method $x_{t+1} = x_t - \alpha_t \, \mathrm{sign}(g_t)$, which also received significant attention in the literature [35, 2, 45, 23].

**BCOS-m**  Using the stochastic momentum as search direction has a long history in stochastic approximation [e.g., 18, 34, 40]. It has become the default option for modern deep learning due to its superior performance compared with using plain stochastic gradients. Following the standard notation in machine learning, we use $m_t$ to denote the momentum, as shown in Algorithm 2. We call it BCOS-m to signify the use of momentum as the search direction. BCOS-m employs a second smoothing factor $\beta_2$ to calculate the EMA of $m_t^2$. These two smoothing factors $\beta_1$ and $\beta_2$ do not need to be the same and can be chosen independently in practice.

We notice that BCOS-m is very similar to Adam as given in (5) and (6). The difference is that in Adam, $v_t$ is the EMA of $g_t^2$ instead of $m_t^2$. From BCOS perspective, Adam has a mismatch between the search direction $m_t$ and the second moment estimator based on $g_t^2$, which must be compensated for by a larger smoothing factor $\beta_2$ (because $m_t$ itself is a smoothed version of $g_t$). For BCOS-m, using $\beta_2 = \beta_1$ produces as good performance as Adam with the best tuned $\beta_2$ (see Section 5).

### 3.2  BCOS with conditional estimators

Recall that the optimal stepsizes $\widehat{\gamma}_{t,k}$ in (9) and their simplifications $\widetilde{\gamma}_{t,k}$ in (11) and (15) are all based on *conditional* expectation. In Section 3.1, we used coordinate-wise EMA of $d_t^2$ to approximate the conditional expectation $\mathbf{E}_t[d_t^2]$, i.e., $v_t$ as estimator of $\mathbf{E}_t[d_t^2]$ in BCOS-g and of $\mathbf{E}_t[m_t^2]$ in BCOS-m, respectively. In this section, we show that with $m_t$ as the search direction, we can exploit its update form to derive effective *conditional estimators* that can avoid using EMA.

We first repeat the definition of momentum here: $m_t = \beta m_{t-1} + (1-\beta) g_t$ with $\beta \in [0,1)$. To derive an estimator of $\mathbf{E}_t[m_t^2]$, we expand the square and take expectation of each term:

$$
\begin{aligned}
\mathbf{E}_t\big[m_t^2\big] &= \mathbf{E}_t\big[(\beta m_{t-1} + (1-\beta) g_t)^2\big] \\
&= \beta^2 \mathbf{E}_t[m_{t-1}^2] + 2\beta(1-\beta)\mathbf{E}_t[m_{t-1} \odot g_t] + (1-\beta)^2 \mathbf{E}_t[g_t^2] \\
&= \beta^2 m_{t-1}^2 + 2\beta(1-\beta) m_{t-1} \odot \mathbf{E}_t[g_t] + (1-\beta)^2 \mathbf{E}_t[g_t^2],
\end{aligned} \tag{19}
$$

where we used the fact $\mathbf{E}_t[m_{t-1}^2] = m_{t-1}^2$ and $\mathbf{E}_t[m_{t-1}] = m_{t-1}$ thanks to the definition of $\mathbf{E}_t[\cdot]$ in (7). It remains to approximate $\mathbf{E}_t[g_t]$ and $\mathbf{E}_t[g_t^2]$. Clearly a good estimator for $\mathbf{E}_t[g_t]$ is $m_t$. To approximate $\mathbf{E}_t[g_t^2]$, we could use a separate EMA estimator $v_t' = \beta' v_{t-1}' + (1-\beta') g_t^2$, but this introduces another algorithmic state $v_t'$ and a second smoothing factor $\beta'$. Meanwhile, we notice that the factor $(1-\beta)^2$ multiplying $\mathbf{E}_t[g_t^2]$ is usually very small, especially for $\beta$ close to 1. As a result, any error in approximating $\mathbf{E}_t[g_t^2]$ is attenuated by a very small factor, so it may not cause much

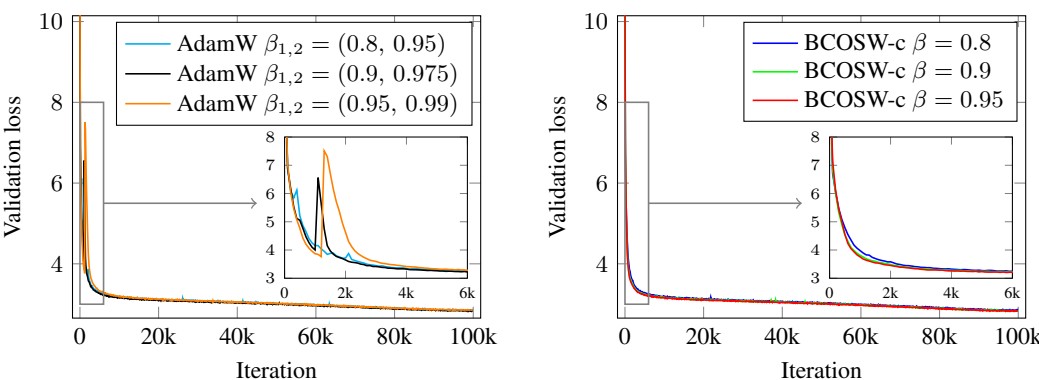

Figure 1: Comparing AdamW and BCOSW-c with different momentum parameters.

difference. Therefore, for simplicity, we choose to approximate $\mathbf{E}_t[g_t^2]$ with $g_t^2$ itself. Combining with approximating $\mathbf{E}_t[g_t]$ with $m_t$, we arrive at the following *conditional* estimator for $\mathbf{E}_t[m_t^2]$:

$$v_t = \beta^2 m_{t-1}^2 + 2\beta(1-\beta)m_{t-1} \odot m_t + (1-\beta)^2 g_t^2. \tag{20}$$

While this can be a very effective estimator, we derive another one that is much simpler and as effective. The key is to approximate $\mathbf{E}[g_t]$ in (19) with $m_{t-1}$ instead of $m_t$, which results in

$$v_t = \beta^2 m_{t-1}^2 + 2\beta(1-\beta)m_{t-1}^2 + (1-\beta)^2 g_t^2$$
$$= \left(1 - (1-\beta)^2\right) m_{t-1}^2 + (1-\beta)^2 g_t^2. \tag{21}$$

It *resembles* the standard EMA estimator in Adam, shown in (6), with an effective smoothing factor

$$\beta' = 1 - (1-\beta)^2,$$

*but with $v_{t-1}$ replaced by $m_{t-1}^2$.* As a result, the estimator in (21) does not need to store $v_{t-1}$, thus requiring fewer optimizer states. This also explains that the second smoothing factor in Adam, $\beta_2$, corresponding to $\beta'$ here, should be much larger or closer to 1 than $\beta$. Specifically, $\beta = 0.9$ roughly corresponds to $\beta' = 0.99$. The estimator in (21) eliminates $\beta_2$ as a second hyper-parameter.

Finally, replacing $v_t$ in BCOS-m with the one in (21) produces Algorithm 3. We call it BCOS-c to signify the *conditional* estimator. It has fewer optimizer states and fewer hyper-parameters to tune.

## 3.3 BCOS with decoupled weight decay

Weight decay is a common practice in training deep learning models to obtain better generalization performance. It can be understood as adding an $L_2$ regularization to the loss function, i.e., minimizing the regularized loss $\mathbf{E}_\xi[f(x,\xi)] + \frac{\lambda}{2}\|x\|^2$. Effectively, the stochastic gradient at $x_t$ becomes $\nabla f(x_t, \xi_t) + \lambda x_t$. We can apply the BCOS family of algorithms by simply replacing $g_t = \nabla f(x_t, \xi_t)$ with $g_t = \nabla f(x_t, \xi_t) + \lambda x_t$. But a more effective way is to use *decoupled weight decay* as proposed in the AdamW algorithm [31]. Specifically, we apply weight decay separately in the BCOS update:

$$x_{t+1} = x_t - \gamma \odot d_t - \alpha_t \lambda x_t = (1 - \alpha_t \lambda)x_t - \gamma \odot d_t.$$

We call the resulting method BCOSW following the naming convention of AdamW. Algorithm 4 shows BCOSW with the conditional estimator. Other variants (-g and -m) can be obtained similarly. A PyTorch implementation of all BCOS and BCOSW variants is given in Appendix A.

## 4 Convergence analysis

In this section, we present the convergence analysis of BCOS and BCOSW. Due to space limit, we focus on BCOSW and give comments on BCOS wherever apply. Our analysis consists of two stages. First, we analyze the convergence properties of the *conceptual* BCOSW method

$$x_{t+1} = (1 - \alpha_t \lambda)x_t - \widetilde{\gamma}_t \odot d_t, \qquad \text{where} \qquad \widetilde{\gamma}_t = \alpha_t \frac{1}{\sqrt{\mathbf{E}_t[d_t^2]}}. \tag{22}$$

It is called "conceptual" because we cannot compute $\mathbf{E}_t[d_t^2]$ exactly in practice. Then, for the practical BCOSW algorithm with stepsize $\gamma_t$ in (18), we bound the difference between the expected steps $\mathbf{E}_t[\gamma_t \odot d_t]$ and $\mathbf{E}_t[\widetilde{\gamma}_t \odot d_t] = \widetilde{\gamma}_t \odot \mathbf{E}_t[d_t]$, which produces the desired convergence guarantee.

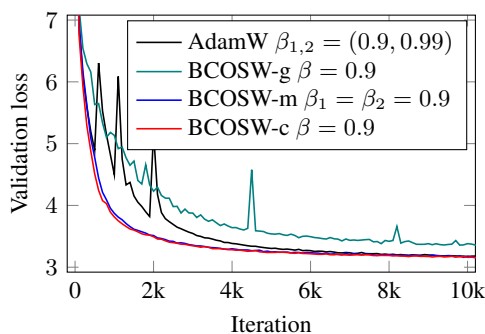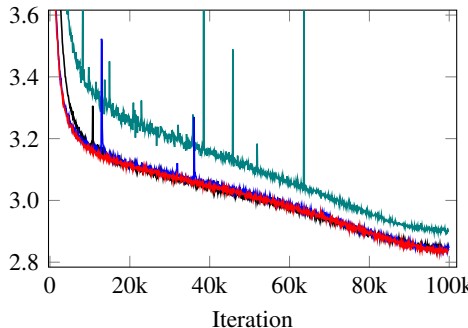

Figure 2: Comparing AdamW and BCOSW. Left: first 10k iterations; Right: all 100k iterations.

First, we need an appropriate condition to build our analysis. For the algorithm $x_{t+1} = x_t - \widetilde{\gamma}_t \odot d_t$, the next iterate $x_{t+1}$ moves closer to $x_*$ in expectation if the expected direction $-\mathbf{E}_t[\widetilde{\gamma}_t \odot d_t]$ aims towards $x_*$ and $\alpha_t$ (a scalar) is sufficiently small. For the conceptual BCOS method, we have

$$\mathbf{E}_t[\widetilde{\gamma}_t \odot d_t] = \mathbf{E}_t\left[\alpha_t \frac{d_t}{\sqrt{\mathbf{E}_t[d_t^2]}}\right] = \alpha_t \frac{\mathbf{E}_t[d_t]}{\sqrt{\mathbf{E}_t[d_t^2]}} = \alpha_t \sqrt{\frac{\mathbf{E}_t[d_t]^2}{\mathbf{E}_t[d_t^2]}} \mathrm{sign}(\mathbf{E}_t[d_t]),$$

where $\mathrm{sign}(\cdot)$ denotes element-wise sign function. Recall the definition of SiF in (14). With single coordinate blocks, we can write the vector of coordinate-wise SiFs as $\rho_t = \frac{\mathbf{E}_t[d_t]^2}{\mathbf{E}_t[d_t^2]} \in [0,1]^n$. Then we have the expected update direction $\mathbf{E}_t[\widetilde{\gamma}_t \odot d_t] = \alpha_t \sqrt{\rho_t} \odot \mathrm{sign}(\mathbf{E}_t[d_t])$. Since $\alpha_t > 0$ is a scalar, we omit it from the statement of the aiming condition below.

**Assumption A** (Aiming condition). *There exists $x_* \in \mathbf{R}^n$ such that*

$$\left\langle x_t - x_*, \ \sqrt{\rho_t} \odot \mathrm{sign}(\mathbf{E}_t[d_t]) + \lambda x_t \right\rangle \geq \lambda \|x_t - x_*\|^2 \tag{23}$$

*holds for all $t \geq 0$ almost surely. If $d_t$ is independent of the past trajectory conditioned on $x_t$, i.e., $\mathbf{E}_t[d_t] = \mathbf{E}[d_t|x_t]$, then it suffices to have (23) hold for every $x \in \mathbf{R}^n$ (independent of the trajectory).*

Notice that we have $\mathbf{E}_t[d_t] = \mathbf{E}[d_t|x_t]$ when, e.g., $d_t = \nabla f(x_t, \xi_t)$ and $\xi_t$ is independent of $x_t$. The aiming conditions assume neither convexity nor smoothness, but it has some overlapping characteristics with convexity, which we discuss in Appendix B.

### 4.1 Analysis of conceptual BCOSW

Our first result concerns the one-step contraction property of the conceptual algorithm in (22).

**Lemma 4.1.** *Suppose Assumption A holds, $\alpha_t \geq 0$ and $\alpha_t \lambda < 1$. Then we have*

$$\mathbf{E}_t\left[\|x_{t+1} - x_*\|^2\right] \leq (1 - \alpha_t \lambda)^2 \|x_t - x_*\|^2 + \alpha_t^2 c_*, \tag{24}$$

*where $c_* = n + \lambda^2 \|x_*\|^2 + 2\lambda \|x_*\|_1$. Thus for sufficiently small $\alpha_t$, $\mathbf{E}_t\left[\|x_{t+1} - x_*\|^2\right] \leq \|x_t - x_*\|^2$.*

In fact, we can prove the following much stronger result of almost sure (a.s.) convergence.

**Theorem 4.1.** *Suppose the stepsize schedule $\{\alpha_t\}_{t\geq 0}$ and weight decay parameter $\lambda$ satisfy*

$$\alpha_t \geq 0, \quad 0 \leq \alpha_t \lambda \leq 1, \quad \forall t \geq 0, \qquad \text{and} \qquad \sum_{t=0}^{\infty} \alpha_t = \infty, \qquad \sum_{t=0}^{\infty} \alpha_t^2 < \infty. \tag{25}$$

*Then Assumption A implies $\|x_t - x_*\| \to 0$ a.s. for the conceptual BCOSW method (22).*

In terms of convergence rate, we can readily obtain linear convergence to a neighborhood of $x_*$ with a constant $\alpha_t$ based on (24). In addition, we have the following result on sublinear convergence.

**Theorem 4.2.** *Consider the conceptual BCOSW method (22) with the stepsize schedule $\alpha_t = \frac{\alpha}{t+1}$ where $1/2 < \alpha\lambda < 1$ is satisfied. Then Assumption A implies that for all $t \geq 1$,*

$$\mathbf{E}[\|x_t - x_*\|^2] \leq \frac{\alpha^2\left(c_* + \lambda^2 \mathbf{E}[\|x_0 - x_*\|^2] + \pi^2 \alpha^2 \lambda^2 c_*/6\right)}{2\alpha\lambda - 1} \frac{1}{t} + \mathcal{O}\left(\frac{1}{t^2} + \frac{1}{t^{2\alpha\lambda}}\right).$$

Without decoupled weight decay, BCOS may also have almost-sure convergence if the aiming condition with $\lambda = 0$ holds with strict inequality for $x_t \neq x_*$. However, the $\mathcal{O}(1/t)$ convergence rate no longer holds. The proofs of the above results are given in Appendix C.

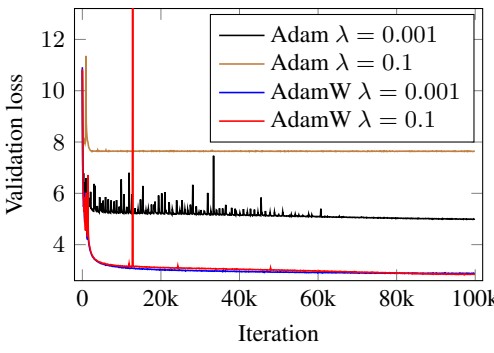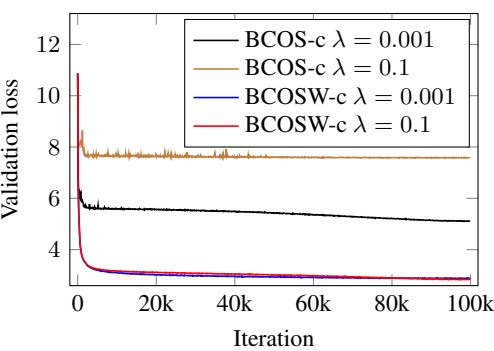

Figure 3: Left: Adam/ AdamW with $\beta_{1,2} = (0.9, 0.99)$. Right: BCOS/BCOSW with $\beta = 0.9$.

## 4.2 Analysis of practical BCOSW

Now we consider the practical BCOSW method $x_{t+1} = (1 - \alpha_t\lambda)x_t - \gamma_t \odot d_t$ with the stepsize vector $\gamma_t$ given in (18). Our analysis is based on bounding the difference between the expected practical update $\mathbf{E}_t[\gamma_t \odot d_t]$ and the expected conceptual update $\mathbf{E}_t[\widetilde{\gamma}_t \odot d_t]$. Intuitively, it boils down to the quality of the estimator $v_t$. Specifically, we need the following assumption on its bias.

**Assumption B.** *There exists $\tau > 0$ and $\epsilon > 0$ such that for all $t \geq 0$ it holds that*

$$\left|\mathbf{E}_t[v_t] - \mathbf{E}_t[d_t^2]\right| \leq \tau\mathbf{E}_t[d_t^2] + \epsilon. \tag{26}$$

Based on this assumption, we have the following bound on the expected update directions.

**Lemma 4.2.** *Under Assumptions B, we have the following bound at each iteration $t$:*

$$\left|\frac{\mathbf{E}_t[d_t]}{\sqrt{\mathbf{E}_t[d_t^2]}} - \mathbf{E}_t\left[\frac{d_t}{\sqrt{v_t + \epsilon}}\right]\right| \leq c_t\left|\frac{\mathbf{E}_t[d_t]}{\sqrt{\mathbf{E}_t[d_t^2]}}\right| + \mathcal{O}(\epsilon) + \mathcal{O}(\mathrm{Var}_t(v_t)), \tag{27}$$

*where $\mathcal{O}(\mathrm{Var}_t(v_t))$ includes terms such as $\mathbf{E}_t[(d_t - \mathbf{E}_t[d_t])(v_t - \mathbf{E}_t[v_t])^2]$ and $\mathbf{E}_t[(v_t - \mathbf{E}_t[v_t])^3]$ and higher-order terms. The coefficient $c_t$ is defined as*

$$c_t := \frac{4\tau + 3\tau^2}{8} + \frac{8 + 4\tau + 3\tau^2}{16}\left(\frac{1}{\mathrm{SNR}_t(v_t + \epsilon)} + \frac{1}{\sqrt{\mathrm{SNR}_t(d_t)}\sqrt{\mathrm{SNR}_t(v_t + \epsilon)}}\right). \tag{28}$$

Here, $\mathrm{SNR}_t(\cdot)$ denotes *conditional Signal-to-Noise Ratio*. Specifically, $\mathrm{SNR}_t(d_t) = \frac{\mathbf{E}_t[d_t]^2}{\mathrm{Var}_t(d_t)} = \frac{\rho_t}{1 - \rho_t}$ and $\mathrm{SNR}_t(v_t + \epsilon) = \frac{\mathbf{E}[v_t + \epsilon]^2}{\mathrm{Var}_t(v_t + \epsilon)} = \frac{\mathbf{E}[v_t + \epsilon]^2}{\mathrm{Var}_t(v_t)}$. This leads to the following result for practical BCOSW:

**Theorem 4.3.** *Suppose Assumptions A and B holds, $\{\alpha_t\}$ satisfies (25) and $\|d_t\|$ is bounded almost surely. Let $\delta$ be the smallest constant such that, for all $t \geq 0$,*

$$2c_t\|\sqrt{\rho_t}\| + \mathcal{O}(\epsilon) + \mathcal{O}(\mathrm{Var}_t(v_t)) \leq \lambda\delta. \tag{29}$$

*Then we have $\limsup_{t\to\infty}\|x_t - x_*\|^2 \leq \delta^2$, meaning a.s. convergence to a neighborhood of $x_*$.*

In fact, it is sufficient for $\lambda\delta$ to be the $\limsup_{\to\infty}$ of the left-hand side of (29) (see Appendix D.2). We notice from (28) that $c_t$ is small if the estimator $v_t$ has low bias (small $\tau$) and low variance (high SNR). In addition, it also helps to have high SNR of $d_t$, for example, by using $m_t$ rather than $g_t$.

Let's examine the bias-variance trade-off of the effective estimator $v_t$ used by popular optimizers:

- The classical SGD method (with $d_t = g_t$ or $d_t = m_t$) effectively uses a constant $v_t$, which has zero variance but high bias $|\mathbf{E}_t[v_t] - \mathbf{E}_t[d_t^2]| = |v - \mathbf{E}_t[d_t^2]|$ for some constant $v$.
- Sign-SGD effectively uses $v_t = d_t^2$, which has no bias but high variance $\mathrm{Var}_t(v_t) = \mathrm{Var}_t(d_t)$.
- The conditional estimator of BCOS-c has the following bias and variance (see Appendix E)
$$\mathbf{E}_t[v_t] - \mathbf{E}_t[d_t^2] = 2\beta(1 - \beta)m_{t-1}\big(m_{t-1} - \mathbf{E}_t[g_t]\big), \qquad \mathrm{Var}_t(v_t) = (1 - \beta)^4\mathrm{Var}_t(g_t^2).$$
  Its bias is a small fraction of the bias of $m_{t-1}$ and it has a very small variance.
- For Adam, we do not have a simple expression for its bias, but $\mathrm{Var}_t(v_t) = (1 - \beta_2)^2\mathrm{Var}_t(m_t^2)$.

In summary, our convergence analysis can be applied to a variety of different optimizers, including Adam and AdamW, by characterizing their bias-variance trade-off (see Appendix E).

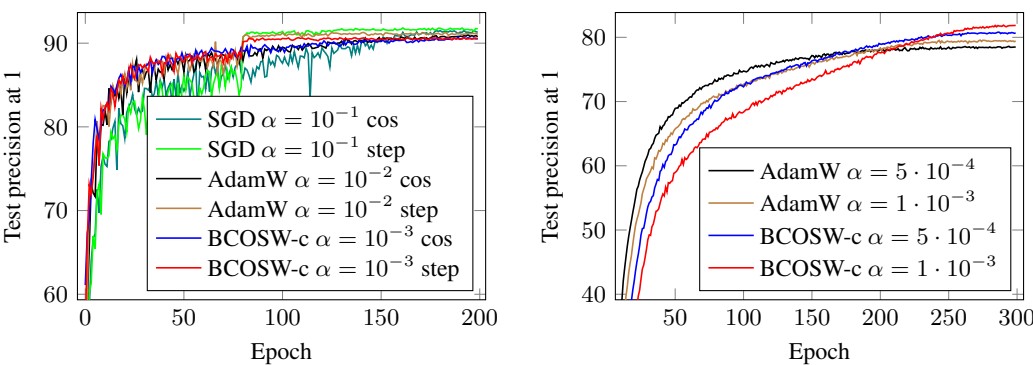

Figure 4: Left: ResNet-20 on CIFAR10. Right: Vision Transformer on ImageNet.

## 5 Numerical experiments

We present preliminary experiments to compare BCOS with Adam, specifically their variants with decoupled weight decay. Among the BCOSW family, we focus on BCOSW-c (Algorithm 4).

Our first set of experiments are conducted on training the small GPT2 model with 124 million parameters [36] on the OpenWebText dataset [16]. We use global batch size 512 and run all experiments for 100k iterations with the first 2k for linear warmup and then cosine decay on $\{\alpha_t\}$. The default hyper-parameters are chosen (based on a coarse sweep) as: peak stepsize $\alpha_{\max} = 0.002$, final stepsize $\alpha_{\min} = 0.01\alpha_{\max}$, $\epsilon = 10^{-6}$ and weight decay $\lambda = 0.1$.

Figure 1 (left) shows the test loss of AdamW with different combination of $\beta_1$ and $\beta_2$. For each value of $\beta_1 \in \{0.8, 0.9, 0.95\}$, we choose the best $\beta_2$ after sweeping $\beta_2 \in \{0.8, 0.9, 0.95, 0.975, 0.99\}$. Their final loss achieved are all very close around 2.82. For most $(\beta_1, \beta_2)$ combinations, we observe loss spikes, especially at the beginning of the training (as shown in the inset). In contrast, Figure 1 (right) shows that BCOSW-c obtains the same final loss but with very smooth loss curve.

Figure 2 compares the test loss of AdamW against the three variants BCOSW-g, -m, and -c. We observe that BCOSW-g is significantly worse than the momentum-based methods. The loss curves for the momentum-based methods are all very close, but with spikes for both AdamW and BCOSW-m.

Figure 3 illustrates the difference between algoritms with and without decoupled weight decay. BCOS-c converges to much higher loss than BCOSW-c, and different values of $\lambda$ (weight decay) makes dramatic difference for BCOS-c but cause little change to BCOSW-c. The same phenomenon happens for Adam versus AdamW, and we again observe spikes from their loss curve.

Finally, in Figure 4, we compare different algorithms for training ResNet-20 [20] on the CIFAR10 dataset [27], and also training the Vision Transformer (ViT) [50] on the ImageNet dataset [11]. For the ResNet task, we tried both cosine decay (drop by factor 100) and step decay (drop by 10 at epochs 80, 120, 150). The hyper-parameters chosen are: $\beta = 0.9$ for SGD and BCOSW-c, and $\beta_{1,2} = (0.9, 0.99)$ for AdamW. We observe that the best-performing stepsize schedules are quite different for different methods. This prompt the need of tuning hyper-parameters for BCOSW for different tasks even though it shares similar tuned hyper-parameters as AdamW on the GPT2 task.

For the ViT task, although the best tuned stepsize schedules are similar between AdamW and BCOSW, their training and test curves look quite different. Figure 4 (right) shows that the test precision curves for BCOSW-c raises slowly but reaches slightly higher precision at the end.

These preliminary experiments demonstrate that BCOSW-c can obtain competitive performance compared with the state-of-the-art method AdamW, but with fewer optimizer states and fewer hyper-parameters to tune. We are conducting additional empirical study to fully understand its potential.

## 6 Conclusion

BCOS is a stochastic approximation method that exploits the flexibility of taking different coordinate-wise stepsizes. Rather than using sophisticated ideas from optimization such as preconditioning, it builds upon the simple idea of coordinate-wise contraction and focuses on constructing efficient statistical estimators, especially through conditional expectation, in determining the stepsizes.

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

# A  PyTorch implementation of BCOS

Listing 1: BCOS and BCOSW implementation as a single PyTorch Optimizer

```python
import torch
from torch.optim import Optimizer

class BCOS_short(Optimizer):
    def __init__(self, params, lr, beta=0.9, eps=1e-6,
                 weight_decay=0.1, mode='c', decouple_wd=True):

        defaults = dict(lr=lr, beta=beta, eps=eps, wd=weight_decay)
        super().__init__(params, defaults)

        if mode not in ['g', 'm', 'c']:
            raise ValueError(f"BCOS mode {mode} not supported")
        self.mode = mode
        self.decouple_wd = decouple_wd        # True for BCOSW

    def step(self, closure = None):

        for group in self.param_groups:
            lr = group["lr"]
            beta = group["beta"]
            eps = group["eps"]
            wd = group["wd"]

            for p in group["params"]:
                if not p.requires_grad:
                    continue

                state = self.state[p]
                g = p.grad

                # initialize optimizer states for specific modes
                if self.mode in ['m', 'c'] and 'm' not in state:
                    state['m'] = g.detach().clone()
                if self.mode in ['g', 'm'] and 'v' not in state:
                    state['v'] = g.detach().square()

                # decoupled weight decay or absorb in gradient
                if self.decouple_wd:     # p := (1 - lr * wd) * p
                    p.data.mul_(1 - lr * wd)
                else:                          # g := g + wd * p
                    g.data.add_(p.data, alpha = wd)

                if self.mode in ['m', 'c']:
                    m = state['m']
                    if self.mode == 'c':
                        beta_v = 1 - (1 - beta)**2
                        g2 = g.detach().square()
                        v = beta_v * m.square() + (1 - beta_v) * g2
                    # update momentum
                    m.mul_(beta).add_(g.detach(), alpha=1 - beta)
                    d = m
                else:
                    d = g.detach()

                if self.mode in ['g', 'm']:      # EMA estimator
                    v = state['v']
                    v.mul_(beta).add_(d.square(), alpha=1 - beta)

                # BCOS update: p := p - lr * (d / (sqrt(v) + eps))
                p.data.add_(d.div(v.sqrt() + eps), alpha= - lr)
```

## B  Aiming condition and convexity

In the paper we have focused on the special case of single coordinate blocks. To investigate the relation between the aiming condition and convexity, it is more instructive to examine the general block structure. For general block partitions $\cup_{k=1}^m \mathcal{I}_k$, employing a block-coordinate stepsize vector $s_t$ where each block $\mathcal{I}_k$ of $s_t$ is defined as $s_{t,k} = \widetilde{\gamma}_{t,k} \mathbf{1}_{n_k}$ yields iterative methods of the form

$$x_{t+1} = x_t - s_t \odot d_t, \tag{30}$$

with conceptual BCOS stepsizes

$$\widetilde{\gamma}_{t,k} = \frac{1}{\sqrt{\mathbf{E}_t\left[\|d_{t,k}\|^2\right]}}, \qquad k = 1, \ldots, m.$$

The corresponding aiming condition is as follows, which guarantees one-step improvement.

**Assumption C.** *There exists $x_* \in \mathbf{R}^n$ such that*

$$\sum_{k=1}^m \left\langle x_t - x_*, \frac{\mathbf{E}_t[d_t]}{\sqrt{\mathbf{E}_t\left[\|d_{t,k}\|^2\right]}} \right\rangle \geq 0, \tag{31}$$

*holds for all $t \geq 0$ almost surely. If $d_t$ is independent of the past trajectory conditioned on $x_t$, i.e., $\mathbf{E}_t[d_t] = \mathbf{E}[d_t|x_t]$, then it suffices to have (31) hold for every $x \in \mathbf{R}^n$.*

Assumption C allows us to conduct a comparative analysis of the aiming condition and the classical convexity assumption, highlighting their similarities and key differences. For the sake of simplicity in our exposition, we will assume that the stochastic search direction $d_t$ is trajectory independent, i.e., $\mathbf{E}_t[d_t] = \mathbf{E}[d_t|x_t]$, allowing us to drop the subscript $t$. We further assume that $d_t$ satisfies $\mathbf{E}[d] = \nabla f(x)$. Simplifying (31):

$$\sum_{k=1}^m \left\langle x_k - x_{*,k}, \frac{\nabla f(x)_k}{\|\nabla f(x)_k\|} \right\rangle \geq 0, \qquad \forall x. \tag{32}$$

In the specific case of a full-dimensional block stepsize, where $\widetilde{\gamma}_t = \frac{1}{\|\nabla f(x)\|} \in \mathbf{R}_+$ is a scaler and the stepsize vector is $s_t = \widetilde{\gamma}_t \mathbf{1}_n$, the aiming condition simplifies to:

$$\langle x - x_*, \nabla f(x) \rangle \geq 0, \qquad \forall x. \tag{33}$$

Condition (33) is directly implied by the classical convex assumption, which states:

$$\langle x - y, \nabla f(x) - \nabla f(y) \rangle \geq 0, \qquad \forall x, y. \tag{34}$$

To see the implication, simply substitute $y = x_*$ and $\nabla f(x_*) = 0$ into the above convex inequality.

However, the aiming condition under a general block partition exhibits a significant departure from the classical notion of convexity, as expected update directions deviate from true gradients and become axis-aligned. Consider the extreme case of coordinate-wise stepsizes, where $s_t = \widetilde{\gamma}_t \in \mathbf{R}^n$ and each element is chosen as $\widetilde{\gamma}_{t,k} = \frac{1}{\sqrt{\nabla f(x_k)^2}} = \frac{1}{|\nabla f(x_k)|}$. The specific choice of stepsize yields an aiming condition of the form:

$$\langle x - x_*, \text{sign}(\nabla f(x)) \rangle \geq 0, \qquad \forall x. \tag{35}$$

To illustrate the fundamental differences between this coordinate-wise aiming condition (35) and the standard convexity assumption (34), we provide the following two counterexamples, each satisfying one condition while failing the other:

- *Aiming but not convex*: Let $f(x) := \log(x)$ with the optimal solution $x_* = 0$. On the domain of $\mathbf{R}_+$, the gradient is $f''(x) = \frac{1}{x}$, and thus $\text{sign}(f'(x)) = 1$ for all $x > 0$. Consequently, for any $x \in \mathbf{R}_+$, we have

$$\langle x - x_*, \text{sign}(\nabla f(x)) \rangle = x \geq 0,$$

  satisfying the aiming condition (35). However, $\log(x)$ is a concave function, thus failing the convex inequality (34).

- *Convex but not aiming*: Consider the quadratic function class $f : \mathbf{R}^2 \to \mathbf{R}$, $f(x) = \frac{1}{2}x^T Ax$. Choose coefficient matrix $A$:

$$A = \begin{pmatrix} 1 \\ -2 \end{pmatrix}\begin{pmatrix} 1 \\ -2 \end{pmatrix}^T = \begin{bmatrix} 1 & -2 \\ -2 & 4 \end{bmatrix} \succeq 0.$$

Since $A$ is positive semidefinite, the function $f$ is convex and attains its minimum at $x_* = \mathbf{0}$. The gradient of $f$ is

$$\nabla f(x) = Ax = \begin{bmatrix} x_1 - 2x_2 \\ -2x_1 + 4x_2. \end{bmatrix}.$$

Evaluating the aiming condition (35) at $x = (1.5, 1)^T$, we get

$$\langle x - x_*, \operatorname{sign}(\nabla f(x))\rangle = 1.5 \times \operatorname{sign}(-0.5) + 1 \times \operatorname{sign}(1) = 1.5 \times (-1) + 1 \times 1 = -0.5 \leq 0.$$

Thus, the aiming condition (35) at this point even though $f$ is convex.

## C    Convergence analysis of conceptual BCOSW

First, we notice that the aiming condition is Assumption A is equivalent to

$$\langle x_t - x_*, \ \sqrt{\rho_t} \odot \operatorname{sign}(\mathbf{E}_t[d_t]) + \lambda x_*\rangle \geq 0, \tag{36}$$

because

$$\langle x_t - x_*, \ \sqrt{\rho_t} \odot \operatorname{sign}(\mathbf{E}_t[d_t]) + \lambda x_*\rangle$$
$$=\langle x_t - x_*, \ \sqrt{\rho_t} \odot \operatorname{sign}(\mathbf{E}_t[d_t]) + \lambda x_t - \lambda x_t + \lambda x_*\rangle$$
$$=\langle x_t - x_*, \ \sqrt{\rho_t} \odot \operatorname{sign}(\mathbf{E}_t[d_t]) + \lambda x_t\rangle - \lambda\|x_t - x_*\|^2.$$

We use it to prove Lemma 4.1.

*Proof of Lemma 4.1.*  Given $x_{t+1} = x_t - \widetilde{\gamma}_t \odot d_t - \alpha_t x_t$, we have

$$\mathbf{E}_t[\|x_{t+1} - x_*\|^2] = \mathbf{E}_t\left[\left\|x_t - \frac{\alpha_t}{\sqrt{\mathbf{E}_t[d_t^2]}} \odot d_t - \alpha_t \lambda x_t - x_*\right\|^2\right]$$

$$= \mathbf{E}_t\left[\left\|(1 - \alpha_t\lambda)x_t - \frac{\alpha_t}{\sqrt{\mathbf{E}_t[d_t^2]}} \odot d_t - (1 - \alpha_t\lambda)x_* - \alpha_t\lambda x_*\right\|^2\right]$$

$$= \mathbf{E}_t\|(1 - \alpha_t\lambda)(x_t - x_*)\|^2 - 2\mathbf{E}_t\left\langle (1 - \alpha_t\lambda)(x_t - x_*), \frac{\alpha_t}{\sqrt{\mathbf{E}_t[d_t^2]}} \odot d_t + \alpha_t\lambda x_*\right\rangle$$

$$+ \mathbf{E}_t\left\|\frac{\alpha_t}{\sqrt{\mathbf{E}_t[d_t^2]}} \odot d_t + \alpha_t\lambda x_*\right\|^2$$

$$= (1 - \alpha_t\lambda)^2 \|x_t - x_*\|^2 - 2\alpha_t(1 - \alpha_t\lambda)\left\langle x_t - x_*, \frac{\mathbf{E}_t[d_t]}{\sqrt{\mathbf{E}_t[d_t^2]}} + \lambda x_*\right\rangle$$

$$+ \alpha_t^2 \sum_k \left(1 + \lambda^2 x_{*,k}^2 + 2\lambda x_{*,k}\frac{\mathbf{E}_t[d_{t,k}]}{\sqrt{\mathbf{E}_t[d_{t,k}^2]}}\right)$$

$$= (1 - \alpha_t\lambda)^2 \|x_t - x_*\|^2 - 2\alpha_t(1 - \alpha_t\lambda)\langle x_t - x_*, \ \sqrt{\rho_t}\operatorname{sign}(\mathbf{E}_t[d_t]) + \lambda x_*\rangle$$

$$+ \alpha_t^2 \sum_k \left(1 + \lambda^2 x_{*,k}^2 + 2\lambda x_{*,k}\sqrt{\rho_{t,k}}\operatorname{sign}(\mathbf{E}_t[d_{t,k}])\right)$$

Under Assumption A, the aiming condition in (36) implies that the inner product in the last equality above is non-negative. With $\alpha_t \geq 0$ and $\alpha_t\lambda \leq 1$, we can drop the inner product term to obtain

$$\mathbf{E}_t[\|x_{t+1} - x_*\|^2] \leq (1 - \alpha_t\lambda)^2 \|x_t - x_*\|^2 + \alpha_t^2 \sum_k \left(1 + \lambda^2 x_{*,k}^2 + 2\lambda x_{*,k}\sqrt{\rho_{t,k}}\operatorname{sign}(\mathbf{E}_t[d_{t,k}])\right)$$

$$\leq (1 - \alpha_t\lambda)^2 \|x_t - x_*\|^2 + \alpha_t^2(n + \lambda^2 \|x_*\|^2 + 2\lambda \|x_*\|_1)$$

where the last inequality follows from the loose upper bound $\sqrt{\rho_{t,k}}\operatorname{sign}(\mathbf{E}_t[d_{t,k}]) \leq 1$.    $\square$

520 The proof of Theorem 4.1 follows from the following almost supermartingale lemma.

521 **Lemma C.1** ("Almost supermartingale", Theorem 1 [39])**.** *) Let $(\Omega, \mathcal{F}, P)$ be a probability space, and*
522 $\mathcal{F}_0 \subset \mathcal{F}_1 \subset \dots$ *be a sequence of sub-$\sigma$-algebras of $\mathcal{F}$. For each $t$, let $X_t, a_t, b_t, c_t$ be non-negative*
523 $\mathcal{F}_t$*-measurable random variables such that*

$$\mathbf{E}[X_{t+1}|\mathcal{F}_t] \leq X_t(1 + a_t) + b_t - c_t. \tag{37}$$

524 *Given $\sum_{t=0}^{\infty} a_t < \infty$ and $\sum_{t=0}^{\infty} b_t < \infty$, then $\lim_{t \to \infty} X_t$ exists and is finite, and $\sum_{t=0}^{\infty} c_t < \infty$*
525 *almost surely (a.s.).*

526 *Proof of Theorem 4.1.* Define $X_t := \|x_t - x_*\|^2$ and $\mathcal{F}_t$ to be the $\sigma$-algebra generated by
527 $X_0, \cdots, X_t$. Lemma 4.1 implies the following recursive relationship

$$
\begin{aligned}
\mathbf{E}[X_{t+1}|\mathcal{F}_t] = \mathbf{E}_t[\|x_{t+1} - x_*\|^2] \\
&\leq (1 - \alpha_t\lambda)^2 \|x_t - x_*\|^2 + \alpha_t^2 c_* \\
&= (1 + \alpha_t^2\lambda^2) \|x_t - x_*\|^2 + \alpha_t^2 c_* - 2\alpha_t\lambda \|x_t - x_*\|^2 \\
&= (1 + a_t) X_t + b_t - c_t,
\end{aligned}
$$

528 In the form of (37), we have $a_t = \alpha_t^2\lambda^2$, $b_t = \alpha_t^2 c_*$, $c_t = 2\alpha_t\lambda \|x_t - x_*\|^2$. Here, $X_t, a_t, b_t, c_t$ are
529 trivially non-negative, and the squared summable assumption of $\alpha_t$ guarantees:

$$\sum_{t=0}^{\infty} a_t = \sum_{t=0}^{\infty} \alpha_t^2\lambda^2 < \infty, \qquad \sum_{t=0}^{\infty} b_t = \sum_{t=0}^{\infty} \alpha_t^2 c_* < \infty.$$

530 So far, we have verified all the assumptions in Lemma C.1, so we conclude that

$$X_t = \|x_t - x_*\|^2 \to X \quad \text{a.s. for some } X < \infty, \qquad \sum_{t=0}^{\infty} c_t = \sum_{t=0}^{\infty} 2\alpha_t\lambda \|x_t - x_*\|^2 < \infty \quad \text{a.s.}$$

531 This is compatible with $\sum_{t=0}^{\infty} \alpha_t = \infty$ only if

$$\|x_t - x_*\|^2 \to 0 \quad \text{a.s.},$$

532 as desired. □

533 To quantify the convergence rate, we study the upper bound on the expected distance to the optimal
534 solution $\mathbf{E}[\|x_T - x_*\|^2]$, after recursively applying BCOSW for $T$ iterations.

535 **Theorem C.1.** *Suppose Assumption A holds, $\alpha_t \geq 0$ and $\alpha_t\lambda \leq 1$. The expected distance to $x_*$*
536 *admits the following upper bound after $T$ iterations of BCOSW:*

$$\mathbf{E}[\|x_T - x_*\|^2] \leq \prod_{t=0}^{T-1}(1 - \alpha_t\lambda)^2 \mathbf{E}\left[\|x_0 - x_*\|^2\right] + \sum_{t=0}^{T-1}\prod_{t'=t+1}^{T-1}(1 - \alpha_{t'}\lambda)^2\alpha_t^2 c_*, \tag{38}$$

537 *where $c_* := (n + \lambda^2 \|x_*\|^2 + 2\lambda \|x_*\|_1)$ denote the constant residual that depends on $x_*$.*

*Proof.* Taking expectation of the recursive relationship (24) and applying the law of total expectation, we obtain:

$$
\begin{aligned}
\mathbf{E}[\|x_T - x_*\|^2] &= \mathbf{E}\left[\mathbf{E}_{T-1}[\|x_T - x_*\|^2]\right] \\
&\leq \mathbf{E}\left[(1 - \alpha_{T-1}\lambda)^2 \|x_{T-1} - x_*\|^2 + \alpha_{T-1}^2 c_*\right] \\
&= (1 - \alpha_{T-1}\lambda)^2 \mathbf{E}\left[\|x_{T-1} - x_*\|^2\right] + \alpha_{T-1}^2 c_* \\
&= (1 - \alpha_{T-1}\lambda)^2 \mathbf{E}\left[\mathbf{E}_{T-1}[\|x_{T-1} - x_*\|^2]\right] + \alpha_{T-1}^2 c_* \\
&\leq (1 - \alpha_{T-1}\lambda)^2 \mathbf{E}\left[(1 - \alpha_{T-2}\lambda)^2 \|x_{T-2} - x_*\|^2 + \alpha_{T-2}^2 c_*\right] + \alpha_{T-1}^2 c_* \\
&= (1 - \alpha_{T-1}\lambda)^2 (1 - \alpha_{T-2}\lambda)^2 \mathbf{E}\left[\|x_{T-2} - x_*\|^2\right] + ((1 - \alpha_{T-1}\lambda)^2 \alpha_{T-2}^2 + \alpha_{T-1}^2) c_* \\
&\ \ \vdots \\
&\leq \prod_{t=0}^{T-1}(1 - \alpha_t \lambda)^2 \mathbf{E}\left[\|x_0 - x_*\|^2\right] + \sum_{t=0}^{T-1}\prod_{t'=t+1}^{T-1}(1 - \alpha_{t'}\lambda)^2 \alpha_t^2 c_*,
\end{aligned}
$$

as desired. $\qquad\square$

Different choices of stepsize schedule lead to different convergence behaviors. Next, we consider two choices of $\alpha_t$: (i) diminishing learning rates $\alpha_t = \frac{\alpha}{t+1}$, which leads to Theorem 4.2 and (ii) constant learning rates $\alpha_t = \alpha$ which lead to linear convergence to a neighborhood of $x_*$.

The proof of Theorem 4.2 is a direct application of a classical result in the 1954 paper of Chung's [7].

**Lemma C.2** (Chung's lemma, Lemma 1 from [7]). *Suppose that $\{X_t\}$ is a sequence of real numbers such that for $t$,*

$$
X_{t+1} \leq \left(1 - \frac{a}{t}\right) X_t + \frac{b}{t^{p+1}}, \tag{39}
$$

*where $a > p > 0, b > 0$. Then*

$$
X_t \leq \frac{b}{a-p}\frac{1}{t^p} + \mathcal{O}\left(\frac{1}{t^{p+1}} + \frac{1}{t^a}\right).
$$

*Proof of Theorem 4.2.* Taking expectation of both sides of (24) with $\alpha_t = \frac{\alpha}{t+1}$ at iteration $T$, we have

$$
\begin{aligned}
\mathbf{E}&[\|x_T - x_*\|^2] \\
&\leq (1 - \alpha_{T-1}\lambda)^2 \mathbf{E}[\|x_{T-1} - x_*\|^2] + \alpha_{T-1}^2 c_* \\
&= \left(1 - \frac{\alpha\lambda}{T}\right)^2 \mathbf{E}[\|x_{T-1} - x_*\|^2] + \frac{\alpha^2}{T^2}c_* \\
&= \left(1 - \frac{2\alpha\lambda}{T}\right) \mathbf{E}[\|x_{T-1} - x_*\|^2] + \frac{\alpha^2}{T^2}\left(c_* + \lambda^2 \mathbf{E}[\|x_{T-1} - x_*\|^2]\right) \\
&\leq \left(1 - \frac{2\alpha\lambda}{T}\right) \mathbf{E}[\|x_{T-1} - x_*\|^2] + \frac{\alpha^2 c_*}{T^2} \\
&\quad + \frac{\alpha^2 \lambda^2}{T^2}\left(\prod_{t=0}^{T-2}\left(1 - \frac{\alpha\lambda}{t+1}\right)^2 \mathbf{E}\left[\|x_0 - x_*\|^2\right] + \sum_{t=0}^{T-2}\prod_{t'=t+1}^{T-2}\left(1 - \frac{\alpha\lambda}{t'+1}\right)^2 \frac{\alpha^2 c_*}{(t+1)^2}\right),
\end{aligned}
$$

where the last inequality is in light of (38) in Theorem C.1 and $\alpha_t = \frac{\alpha}{t+1}$. Upper bounding $\left(1 - \frac{\alpha\lambda}{t+1}\right)^2$ by 1 yields:

$$
\mathbf{E}[\|x_T - x_*\|^2] \leq \left(1 - \frac{2\alpha\lambda}{T}\right) \mathbf{E}[\|x_{T-1} - x_*\|^2] + \frac{\alpha^2 c_*}{T^2} + \frac{\alpha^2 \lambda^2}{T^2}\left(\mathbf{E}\left[\|x_0 - x_*\|^2\right] + \sum_{t=0}^{T-2}\frac{\alpha^2 c_*}{(t+1)^2}\right).
$$

Further replacing the finite sum $\sum_{t=0}^{T-2} \frac{1}{(t+1)^2} = \sum_{t=1}^{T-1} \frac{1}{t^2}$ by its infinite version $\frac{\pi^2}{6}$, we obtain a recursive relationship in the form of (39):

$$\mathbf{E}[\|x_T - x_*\|^2] \le \left(1 - \frac{2\alpha\lambda}{T}\right)\mathbf{E}[\|x_{T-1} - x_*\|^2] + \frac{\alpha^2 c_*}{T^2} + \frac{\alpha^2\lambda^2}{T^2}\left(\mathbf{E}\left[\|x_0 - x_*\|^2\right] + \frac{\pi^2\alpha^2 c_*}{6}\right),$$

with $X_t = \mathbf{E}[\|x_{t-1} - x_*\|^2], a = 2\alpha\lambda, b = \alpha^2 c_* + \alpha^2\lambda^2\left(\mathbf{E}\left[\|x_0 - x_*\|^2\right] + \frac{\pi^2\alpha^2 c_*}{6}\right)$, and $p = 1$, which satisfies the Chung's assumptions $a > 1 = p > 0, b > 0$ because $\alpha\lambda \in (0.5, 1)$. Lemma C.2 implies

$$\mathbf{E}[\|x_T - x_*\|^2] \le \frac{\alpha^2 c_* + \alpha^2\lambda^2\left(\mathbf{E}\left[\|x_0 - x_*\|^2\right] + \frac{\pi^2\alpha^2 c_*}{6}\right)}{2\alpha\lambda - 1}\frac{1}{T} + \mathcal{O}\left(\frac{1}{T^2} + \frac{1}{T^{2\alpha\lambda}}\right),$$

as desired. □

With a constant stepsize, we obtain linear convergence to a neighborhood of $x_*$, as stated in the following corollary.

**Corollary C.2.** *Fix learning rate schedule $\alpha_t = \alpha$ where $\alpha$ satisfies $\alpha\lambda < 1$. Let $x_t$'s be a sequence generated by applying the conceptual BCOSW. Under Assumption A, the asymptotic expected distance to $x_*$ admits the following upper bound:*

$$\mathbf{E}[\|x_T - x_*\|^2] \le (1-\alpha\lambda)^{2T}\mathbf{E}\left[\|x_0 - x_*\|^2\right] + \frac{\alpha^2 c_*}{1 - (1-\alpha\lambda)^2}. \tag{40}$$

*Proof.* A direct application of Theorem C.1 with $\alpha_t = \alpha$ yields the following upper bound on: $\mathbf{E}[\|x_T - x_*\|^2]$

$$\begin{aligned}
\mathbf{E}[\|x_T - x_*\|^2] &\le (1-\alpha\lambda)^{2T}\mathbf{E}\left[\|x_0 - x_*\|^2\right] + \sum_{t=0}^{T-1}(1-\alpha\lambda)^{2(T-t-1)}\alpha^2 c_* \\
&= (1-\alpha\lambda)^{2T}\mathbf{E}\left[\|x_0 - x_*\|^2\right] + \sum_{t=0}^{T-1}(1-\alpha\lambda)^{2t}\alpha^2 c_* \\
&\le (1-\alpha\lambda)^{2T}\mathbf{E}\left[\|x_0 - x_*\|^2\right] + \frac{\alpha^2 c_*}{1 - (1-\alpha\lambda)^2},
\end{aligned}$$

which decreases exponentially with $T$ and converges to a constant. □

# D   Convergence analysis of practical BCOSW

## D.1   Proof of Lemma 4.2

We first prove Lemma 4.2. To proceed, we decompose the error between the expected search directions into two parts (elementwise inequality between vectors):

$$\left|\frac{\mathbf{E}_t[d_t]}{\sqrt{\mathbf{E}[d_t^2]}} - \mathbf{E}_t\left[\frac{d_t}{\sqrt{v_t + \epsilon}}\right]\right| \le \left|\frac{\mathbf{E}_t[d_t]}{\sqrt{\mathbf{E}[d_t^2]}} - \frac{\mathbf{E}_t[d_t]}{\sqrt{\mathbf{E}_t[v_t] + \epsilon}}\right| + \left|\frac{\mathbf{E}_t[d_t]}{\sqrt{\mathbf{E}_t[v_t] + \epsilon}} - \mathbf{E}_t\left[\frac{d_t}{\sqrt{v_t + \epsilon}}\right]\right|. \tag{41}$$

Under certain assumptions on the quality of the estimator $v_t$, we demonstrate that the practical update approximates the conceptual update in expectation by bounding the two terms on the right-hand side separately.

Assumption B leads to an upper bound for the first error term in (41).

**Lemma D.1.** *Under Assumption B, it holds that:*

$$\left|\frac{\mathbf{E}_t[d_t]}{\sqrt{\mathbf{E}_t[d_t^2]}} - \frac{\mathbf{E}_t[d_t]}{\sqrt{\mathbf{E}_t[v_t] + \epsilon}}\right| \le \frac{4\tau + 3\tau^2}{8}\left|\frac{\mathbf{E}_t[d_t]}{\sqrt{\mathbf{E}_t[d_t^2]}}\right| + \mathcal{O}(\epsilon). \tag{42}$$

*Proof.* The proof leverages the second-order Taylor expansion of $g(y) := \frac{1}{\sqrt{y}}$:

$$g(y + \delta) \approx g(y) + g'(y)\delta + \frac{1}{2}g''(y)\delta^2, \qquad \text{where} \quad g'(y) = -\frac{1}{2y^{3/2}}, \quad g''(y) = \frac{3}{4y^{5/2}}.$$

Applying Taylor expansion at $y := \mathbf{E}_t[d_t^2]$ with $\delta := \mathbf{E}_t[v_t] + \epsilon - \mathbf{E}_t[d_t^2]$ yields the following approximation:

$$\left| \frac{\mathbf{E}_t[d_t]}{\sqrt{\mathbf{E}_t[d_t^2]}} - \frac{\mathbf{E}_t[d_t]}{\sqrt{\mathbf{E}_t[v_t] + \epsilon}} \right|$$

$$= |\mathbf{E}_t[d_t](g(y) - g(y + \delta))|$$

$$\approx \left| \mathbf{E}_t[d_t]\left( g'(y)\delta + \frac{1}{2}g''(y)\delta^2 + \mathcal{O}(\delta^3) \right) \right|$$

$$= \left| \mathbf{E}_t[d_t]\left( -\frac{1}{2\mathbf{E}_t[d_t^2]^{3/2}} \cdot (\mathbf{E}_t[v_t] + \epsilon - \mathbf{E}_t[d_t^2]) + \frac{3}{8\mathbf{E}_t[d_t^2]^{5/2}} \cdot (\mathbf{E}_t[v_t] + \epsilon - \mathbf{E}_t[d_t^2])^2 \right) \right| + \mathcal{O}(\epsilon)$$

$$\leq \left| \mathbf{E}_t[d_t]\left( \frac{1}{2\mathbf{E}_t[d_t^2]^{3/2}} \cdot \tau\mathbf{E}_t[d_t^2] + \frac{3}{8\mathbf{E}_t[d_t^2]^{5/2}} \cdot \tau^2\mathbf{E}_t[d_t^2]^2 \right) \right| + \mathcal{O}(\epsilon) \tag{43}$$

$$\leq \frac{4\tau + 3\tau^2}{8} \left| \frac{\mathbf{E}_t[d_t]}{\sqrt{\mathbf{E}_t[d_t^2]}} \right| + \mathcal{O}(\epsilon),$$

where (43) is a consequence of Assumption B. $\qquad\square$

To establish the upper bound on the second error term in (41), $\left| \frac{\mathbf{E}_t[d_t]}{\sqrt{\mathbf{E}_t[v_t] + \epsilon}} - \mathbf{E}_t\left[ \frac{d_t}{\sqrt{v_t + \epsilon}} \right] \right|$, we present a useful approximation for general differential function $g$.

**Lemma D.2.** *For any differentiable function $g$ and random variable $X \in \mathbf{R}^n$, the following expansion holds:*

$$\mathbf{E}[g(X)] = g(\mathbf{E}[X]) + \frac{1}{2}\langle \nabla^2 g(\mathbf{E}[X]), \mathrm{Cov}(X) \rangle + \sum_{p=3}^{\infty} \frac{D^p g(\mathbf{E}[X])}{p!} \mathbf{E}\left[ (X - \mathbf{E}[X])^p \right], \tag{44}$$

*where $\langle \cdot, \cdot \rangle$ denotes matrix inner product, i.e, $\langle A, B \rangle = \mathrm{Tr}(A^T B)$, and $p \in \mathbf{N}^n$ and*

$$D^p g(\mathbf{E}[X]) = \frac{\partial^{|p|} g}{\partial X^p} = \frac{\partial^{p_1 + \cdots + p_n} g}{\partial X_1^{p_1} \cdots \partial X_n^{p_n}}.$$

*Proof.* Let $\delta := X - \mathbf{E}[X]$. The second-order Taylor expansion of $g$ at $\mathbf{E}[X]$ yields

$$g(X) = g(\mathbf{E}[X]) + \nabla g(\mathbf{E}[X])^T \delta + \frac{1}{2}\delta^T \nabla^2 g(\mathbf{E}[X])\delta + \sum_{p=3}^{\infty} \frac{D^p g(\mathbf{E}[X])}{p!} \delta^p.$$

Taking expectation with respect to $X$, we have

$$\mathbf{E}[g(X)] = g(\mathbf{E}[X]) + \nabla g(\mathbf{E}[X])^T \mathbf{E}[\delta] + \frac{1}{2}\mathbf{E}[\delta^T \nabla^2 g(\mathbf{E}[X])\delta] + \sum_{p=3}^{\infty} \frac{D^p g(\mathbf{E}[X])}{p!} \mathbf{E}[\delta^p]$$

$$= g(\mathbf{E}[X]) + 0 + \frac{1}{2}\langle \nabla^2 g(\mathbf{E}[X]), \mathbf{E}[\delta\delta^T] \rangle + \sum_{p=3}^{\infty} \frac{D^p g(\mathbf{E}[X])}{p!} \mathbf{E}[\delta^p],$$

where $\mathbf{E}[\delta\delta^T] = \mathrm{Cov}(X)$ and $\mathbf{E}[\delta^p] = \mathbf{E}[(X - \mathbf{E}[X])^p]$. $\qquad\square$

The following lemma provides an approximation for $\mathbf{E}[g]$ with $g(Y, Z) := \frac{Y}{\sqrt{Z}}$.

**Lemma D.3.** *Let $Y, Z$ be two random variables and $Z > 0$ almost surely, then*

$$\mathbf{E}\left[ \frac{Y}{\sqrt{Z}} \right] = \frac{\mathbf{E}[Y]}{\sqrt{\mathbf{E}[Z]}}\left( 1 - \frac{\mathrm{Cov}(Y, Z)}{2\mathbf{E}[Y]\mathbf{E}[Z]} + \frac{3\mathrm{Var}(Z)}{8\mathbf{E}[Z]^2} \right)$$
$$+ \mathcal{O}\left( \mathbf{E}[(Y - \mathbf{E}[Y])(Z - \mathbf{E}[Z])^2] \right) + \mathcal{O}\left( \mathbf{E}[(Z - \mathbf{E}[Z])^3] \right). \tag{45}$$

588 *Proof.* We apply Lemma D.2 with $X := (Y, Z)$ and $g(x) = g(y, z) := \frac{y}{\sqrt{z}}$. First, the gradient and
589 Hessian of $g$ can be calculated as

$$\nabla g(x) = \nabla g(y, z) = \begin{pmatrix} \frac{1}{z^{1/2}} \\ -\frac{y}{2z^{3/2}} \end{pmatrix}, \qquad \nabla^2 g(x) = \nabla^2 g(y, z) = \begin{bmatrix} 0, & -\frac{1}{2z^{3/2}} \\ -\frac{1}{2z^{3/2}}, & \frac{3y}{4z^{5/2}} \end{bmatrix}.$$

590 For general $p$-th partial derivative, we derive the following result for any $q \in [0, p]$:

$$\frac{\partial^p g}{\partial y^q \partial z^{p-q}} = \frac{\partial^{p-q}}{\partial z^{p-q}} \left( \frac{\partial^q g}{\partial y^q} \right) = \begin{cases} 0 & \text{if } q \geq 2, \\ \frac{\partial^{p-1}}{\partial z^{p-1}} \frac{1}{\sqrt{z}} = (-1)^{p-1} \frac{(2p-2)!}{4^{p-1}(p-1)!} z^{-\frac{2p-1}{2}} & \text{if } q = 1, \\ y \cdot \frac{\partial^p}{\partial z^p} \frac{1}{\sqrt{z}} = (-1)^p \frac{(2p)!}{4^p p!} y z^{-\frac{2p+1}{2}} & \text{if } q = 0. \end{cases}$$

591 which Substitute the gradient, Hessian and $p$-th order partial derivative into (44), we get

$$\mathbf{E}\left[ \frac{Y}{\sqrt{Z}} \right]$$

$$= \frac{\mathbf{E}[Y]}{\sqrt{\mathbf{E}[Z]}} - \mathbf{E}\left[ \frac{(Y - \mathbf{E}[Y])(Z - \mathbf{E}[Z])}{2\mathbf{E}[Z]^{3/2}} \right] + \mathbf{E}\left[ \frac{3\mathbf{E}[Y](Z - \mathbf{E}[Z])^2}{8\mathbf{E}[Z]^{5/2}} \right]$$

$$+ \sum_{p=3}^{\infty} \frac{1}{p!} \left( \frac{p(2p-2)!}{4^{p-1}(p-1)!} \mathbf{E}\left[ \frac{(Y - \mathbf{E}[Y])(Z - \mathbf{E}[Z])^{p-1}}{\mathbf{E}[Z]^{\frac{2p-1}{2}}} \right] + (-1)^p \frac{(2p)!}{4^p p!} \mathbf{E}\left[ \frac{\mathbf{E}[Y](Z - \mathbf{E}[Z])^p}{\mathbf{E}[Z]^{\frac{2p+1}{2}}} \right] \right)$$

$$= \frac{\mathbf{E}[Y]}{\sqrt{\mathbf{E}[Z]}} - \frac{\mathrm{Cov}(Y, Z)}{2\mathbf{E}[Z]^{3/2}} + \frac{3\mathbf{E}[Y]\mathrm{Var}(Z)}{8\mathbf{E}[Z]^{5/2}} + \mathcal{O}\left( \mathbf{E}[(Y - \mathbf{E}[Y])(Z - \mathbf{E}[Z])^2] \right) + \mathcal{O}\left( \mathbf{E}[(Z - \mathbf{E}[Z])^3] \right)$$

$$= \frac{\mathbf{E}[Y]}{\sqrt{\mathbf{E}[Z]}} \left( 1 - \frac{\mathrm{Cov}(Y, Z)}{2\mathbf{E}[Y]\mathbf{E}[Z]} + \frac{3\mathrm{Var}(Z)}{8\mathbf{E}[Z]^2} \right) + \mathcal{O}\left( \mathbf{E}[(Y - \mathbf{E}[Y])(Z - \mathbf{E}[Z])^2] \right) + \mathcal{O}\left( \mathbf{E}[(Z - \mathbf{E}[Z])^3] \right),$$

592 as desired. $\square$

593 A combination of the consequence of Lemma D.3 and Assumption B culminates in an upper bound
594 on the second error term.

595 **Lemma D.4.** *Define signal-noise-ratio* $\mathrm{SNR}_t(Y) := \frac{\mathbf{E}_t[Y_t]^2}{\mathrm{Var}_t(Y_t)}$. *Under Assumptions B, we have*

$$\left| \frac{\mathbf{E}_t[d_t]}{\sqrt{\mathbf{E}_t[v_t] + \epsilon}} - \mathbf{E}_t\left[ \frac{d_t}{\sqrt{v_t + \epsilon}} \right] \right|$$

$$\leq \left( \frac{8 + 4\tau + 3\tau^2}{8} \right) \left| -\frac{\mathrm{Corr}_t(d_t, v_t + \epsilon)}{2\sqrt{\mathrm{SNR}_t(v_t + \epsilon)}} \sqrt{\frac{1}{\rho_t} - 1} + \frac{3}{8\mathrm{SNR}_t(v_t + \epsilon)} \right| \cdot \left| \frac{\mathbf{E}_t[d_t]}{\sqrt{\mathbf{E}_t[d_t^2]}} \right|$$

$$+ \mathcal{O}(\epsilon) + \mathcal{O}\left( \mathbf{E}_t[(Y - \mathbf{E}_t[Y])(Z - \mathbf{E}_t[Z])^2] \right) + \mathcal{O}\left( \mathbf{E}_t[(Z - \mathbf{E}_t[Z])^3] \right)$$

596 *Proof.* Following Lemma D.3 with $Y := d_t$, $Z := v_t + \epsilon$, we get

$$\left| \frac{\mathbf{E}_t[d_t]}{\sqrt{\mathbf{E}_t[v_t] + \epsilon}} - \mathbf{E}_t\left[ \frac{d_t}{\sqrt{v_t + \epsilon}} \right] \right| \leq \left| \frac{\mathbf{E}_t[d_t]}{\sqrt{\mathbf{E}_t[v_t] + \epsilon}} \right| \cdot \left| -\frac{\mathrm{Cov}_t(d_t, v_t + \epsilon)}{2\mathbf{E}_t[d_t]\mathbf{E}_t[v_t + \epsilon]} + \frac{3\mathrm{Var}_t(v_t + \epsilon)}{8\mathbf{E}_t[v_t + \epsilon]^2} \right| \quad (46)$$

$$+ \mathcal{O}\left( \mathbf{E}_t[(Y - \mathbf{E}_t[Y])(Z - \mathbf{E}_t[Z])^2] \right) + \mathcal{O}\left( \mathbf{E}_t[(Z - \mathbf{E}_t[Z])^3] \right).$$

597 We express the covariance between $d$ and $v + \epsilon$ based on the definitions of SNR as follows:

$$\mathrm{Cov}_t(d_t, v_t + \epsilon) = \mathbf{E}_t[d_t]\mathbf{E}_t[v_t + \epsilon] \cdot \frac{\mathrm{Cov}_t(d_t, v_t + \epsilon)}{\sqrt{\mathrm{Var}_t(d_t)\mathrm{Var}_t(v_t + \epsilon)}} \sqrt{\frac{\mathrm{Var}_t(d_t)\mathrm{Var}_t(v_t + \epsilon)}{\mathbf{E}_t[d_t]^2 \mathbf{E}_t[v_t + \epsilon]^2}}$$

$$= \mathbf{E}_t[d_t]\mathbf{E}_t[v_t + \epsilon] \cdot \frac{\mathrm{Corr}_t(d_t, v_t + \epsilon)}{\sqrt{\mathrm{SNR}_t(d_t)\mathrm{SNR}_t(v_t + \epsilon)}},$$

where $\mathrm{SNR}_t(d_t)$ is closely connected to the signal fraction $\rho_t$, defined as $\rho_t := \frac{\mathbf{E}_t[d_t]^2}{\mathbf{E}_t[d_t^2]}$:

$$\mathrm{SNR}_t(d_t) = \frac{\mathbf{E}_t[d_t]^2}{\mathrm{Var}_t(d_t)} = \frac{1}{\mathrm{Var}_t(d_t)/\mathbf{E}_t[d_t]^2} = \frac{1}{(\mathrm{Var}_t(d_t) + \mathbf{E}_t[d_t]^2)/\mathbf{E}_t[d_t]^2 - 1} = \frac{1}{1/\rho_t - 1}.$$

The first term (46) admits the following upper bound:

$$(46) = \left| \frac{\mathbf{E}_t[d_t]}{\sqrt{\mathbf{E}_t[v_t] + \epsilon}} \right| \cdot \left| -\frac{\mathrm{Cov}_t(d_t, v_t + \epsilon)}{2\mathbf{E}_t[d_t]\mathbf{E}_t[v_t + \epsilon]} + \frac{3\mathrm{Var}_t(v_t + \epsilon)}{8\mathbf{E}_t[v_t + \epsilon]^2} \right|$$

$$= \left| \frac{\mathbf{E}_t[d_t]}{\sqrt{\mathbf{E}_t[v_t] + \epsilon}} \right| \cdot \left| -\frac{\mathbf{E}[d_t]\mathbf{E}[v_t + \epsilon]}{2\mathbf{E}_t[d_t]\mathbf{E}_t[v_t + \epsilon]} \cdot \frac{\mathrm{Corr}_t(d_t, v_t + \epsilon)}{\sqrt{\mathrm{SNR}_t(v_t + \epsilon)}} \sqrt{\frac{1}{\rho_t} - 1} + \frac{3}{8\mathrm{SNR}_t(v_t + \epsilon)} \right|$$

$$= \left| \frac{\mathbf{E}_t[d_t]}{\sqrt{\mathbf{E}_t[v_t] + \epsilon}} \right| \cdot \left| -\frac{\mathrm{Corr}_t(d_t, v_t + \epsilon)}{2\sqrt{\mathrm{SNR}_t(v_t + \epsilon)}} \sqrt{\frac{1}{\rho_t} - 1} + \frac{3}{8\mathrm{SNR}_t(v_t + \epsilon)} \right|$$

$$\leq \frac{8 + 4\tau + 3\tau^2}{8} \left| -\frac{\mathrm{Corr}_t(d_t, v_t + \epsilon)}{2\sqrt{\mathrm{SNR}_t(v_t + \epsilon)}} \sqrt{\frac{1}{\rho_t} - 1} + \frac{3}{8\mathrm{SNR}_t(v_t + \epsilon)} \right| \cdot \left| \frac{\mathbf{E}_t[d_t]}{\sqrt{\mathbf{E}_t[d_t^2]}} \right| + \mathcal{O}(\epsilon) \tag{47}$$

where (47) is given by Lemma D.1. $\qquad\square$

Combining the upper bounds on two terms on the right-hand side of (41), we finally can prove Lemma 4.2

*Proof of Lemma 4.2.* It follows immediately by triangle inequality, Lemma D.1 and Lemma D.4.

$$\left| \frac{\mathbf{E}_t[d_t]}{\sqrt{\mathbf{E}_t[d_t^2]}} - \mathbf{E}_t\left[ \frac{d_t}{\sqrt{v_t + \epsilon}} \right] \right|$$

$$\leq \left| \frac{\mathbf{E}_t[d_t]}{\sqrt{\mathbf{E}_t[d_t^2]}} - \frac{\mathbf{E}_t[d_t]}{\sqrt{\mathbf{E}_t[v_t] + \epsilon}} \right| + \left| \frac{\mathbf{E}_t[d_t]}{\sqrt{\mathbf{E}_t[v_t] + \epsilon}} \mathbf{E}_t\left[ \frac{d_t}{\sqrt{v_t + \epsilon}} \right] \right|$$

$$\leq \frac{4\tau + 3\tau^2}{8} \left| \frac{\mathbf{E}_t[d_t]}{\sqrt{\mathbf{E}_t[d_t^2]}} \right| + \frac{8 + 4\tau + 3\tau^2}{8} \left| \frac{\mathrm{Corr}_t(d_t, v_t + \epsilon)}{2\sqrt{\mathrm{SNR}_t(v_t + \epsilon)}} \sqrt{\frac{1}{\rho_t} - 1} - \frac{3}{8\mathrm{SNR}_t[v_t + \epsilon]} \right| \cdot \left| \frac{\mathbf{E}_t[d_t]}{\sqrt{\mathbf{E}_t[d_t^2]}} \right|$$

$$+ \mathcal{O}(\epsilon) + \mathcal{O}\left( \mathbf{E}_t[(Y - \mathbf{E}_t[Y])(Z - \mathbf{E}_t[Z])^2] \right) + \mathcal{O}\left( \mathbf{E}_t[(Z - \mathbf{E}_t[Z])^3] \right)$$

Finally, bounding $|\mathrm{Corr}_t(d_t, v_t + \epsilon)|$ by 1 and recognizing $\frac{1}{\rho_t} - 1 = \frac{1}{\mathrm{SNR}_t(d_t)}$ give the desired result. $\qquad\square$

## D.2 Proof of Theorem 4.3

To prove Theorem 4.3, we can use a classical result on stochastic approximation originally due to Dvoretzky [15].

**Theorem D.1** (An extension of Dvoretzky's Theorem). *Let $(\Omega = \{\omega\}, \mathcal{F}, P)$ be a probability space. Let $\{x_t\}$ and $\{y_t\}$ be sequences of random variables such that, for all $t \geq 0$,*

$$x_{t+1}(\omega) = T_t\big(x_0(\omega), \ldots, x_t(\omega)\big) + y_t(\omega), \tag{48}$$

*where the transformation $T_t$ satisfy, for any $x_0, \ldots, x_t \in \mathbf{R}^n$,*

$$\big\| T_t(x_0, \ldots, x_t) - x_* \big\|^2 \leq \max\big\{ a_t, (1 + b_t)\|x_t - x_*\|^2 - c_t + d_t \big\} \tag{49}$$

*and the sequences $\{a_t\}$, $\{b_t\}$, $\{c_t\}$ and $\{d_t\}$ are non-negative and satisfy*

$$\lim_{t \to \infty} a_t = a_\infty, \qquad \sum_{t=0}^{\infty} b_t < \infty, \qquad \sum_{t=0}^{\infty} c_t = \infty, \qquad \sum_{t=1}^{\infty} d_t < \infty. \tag{50}$$

*In addition, suppose the following conditions hold with probability one:*

$$\mathbf{E}[\|x_0\|^2] < \infty, \qquad \sum_{t=0}^{\infty} \mathbf{E}[\|y_t\|^2] < \infty, \qquad \mathbf{E}\big[y_t | x_0, \ldots, x_t\big] = 0 \quad \forall\, t \geq 0.$$

*Then we have with probability one,*

$$\limsup_{t \to \infty} \|x_t - x_*\|^2 \leq a_\infty.$$

**Remark.** There are many extensions of Dvoretzky's original results [15]. Theorem D.1 is a minor variation of Venter [51, Theorem 1]. More concretely,

- Theorem 1 of Venter [51] has the sequence $\{a_t\}$ being a constant sequence, i.e., $a_t = a_\infty$ for all $t \geq 0$. The extension to a non-constant sequence $\{a_t\}$ is outlined in the original work of Dvoretzky [15] and admits a simple proof due to Derman and Sacks [12].

- Theorem 1 of Venter [51] does not include the sequence $\{d_t\}$. The extension with $\sum_{t=0}^\infty d_t < \infty$ is straightforward based on a simple argument of Dvoretzky [15].

- More generally, the sequences $\{a_t\}$, $\{b_t\}$, $\{c_t\}$, $\{d_t\}$ can be non-negative measurable functions of $x_0, \ldots, x_t$, and the conclusion of Theorem D.1 holds if $a_\infty$ is an upper bound on $\limsup_{t\to\infty} a_t(x_0, \ldots, x_t)$ uniformly for all sequences $x_0, \ldots, x_t, \ldots$ [12, 39].

We also need the following lemma.

**Lemma D.5.** *Under Assumptions B, it holds that*

$$\mathbf{E}_t\left[\left\langle x_t - x_*, \frac{d_t}{\sqrt{v_t + \epsilon}}\right\rangle\right] \geq \left\langle x_t - x_*, \frac{\mathbf{E}_t[d_t]}{\sqrt{\mathbf{E}_t[d_t^2]}}\right\rangle$$
$$- \|x_t - x_*\|\left(c_t\|\sqrt{\rho_t}\| + \mathcal{O}(\epsilon) + \mathcal{O}(\mathrm{Var}_t(v_t))\right),$$

*where $c_t$ is given by* (28).

*Proof.* Adding and subtracting the term $\frac{\mathbf{E}_t[d_t]}{\sqrt{\mathbf{E}_t[d_t^2]}}$ from the inner product, we obtain:

$$\mathbf{E}_t\left[\left\langle x_t - x_*, \frac{d_t}{\sqrt{v_t + \epsilon}}\right\rangle\right]$$
$$=\mathbf{E}_t\left[\left\langle x_t - x_*, \frac{\mathbf{E}_t[d_t]}{\sqrt{\mathbf{E}_t[d_t^2]}}\right\rangle\right] + \mathbf{E}_t\left[\left\langle x_t - x_*, \frac{d_t}{\sqrt{v_t + \epsilon}} - \frac{\mathbf{E}_t[d_t]}{\sqrt{\mathbf{E}_t[d_t^2]}}\right\rangle\right]$$
$$\geq\mathbf{E}_t\left[\left\langle x_t - x_*, \frac{\mathbf{E}_t[d_t]}{\sqrt{\mathbf{E}_t[d_t^2]}}\right\rangle\right] - \|x_t - x_*\| \cdot \left\|\mathbf{E}_t\left[\frac{d_t}{\sqrt{v_t + \epsilon}}\right] - \frac{\mathbf{E}_t[d_t]}{\sqrt{\mathbf{E}_t[d_t^2]}}\right\|$$
$$\geq \left\langle x_t - x_*, \frac{\mathbf{E}_t[d_t]}{\sqrt{\mathbf{E}_t[d_t^2]}}\right\rangle - \|x_t - x_*\|\left(c_t\left\|\frac{\mathbf{E}_t[d_t]}{\sqrt{\mathbf{E}_t[d_t^2]}}\right\| + \mathcal{O}(\epsilon) + \mathcal{O}(\mathrm{Var}_t(v_t))\right), \quad (51)$$

where the inequality (51) is due to Lemma 4.2. To finish the proof, we recall $\sqrt{\rho_t} = \frac{\mathbf{E}_t[d_t]}{\sqrt{\mathbf{E}_t[d_t^2]}}$. $\qquad\square$

*Proof of Theorem 4.3.* We can write the practical BCOSW algorithm as

$$x_{t+1} = (1 - \alpha_t\lambda)x_t - \alpha_t\frac{d_t}{\sqrt{v_t + \epsilon}}$$
$$= (1 - \alpha_t\lambda)x_t - \alpha_t\mathbf{E}_t\left[\frac{d_t}{\sqrt{v_t + \epsilon}}\right] + \alpha_t\left(\mathbf{E}_t\left[\frac{d_t}{\sqrt{v_t + \epsilon}}\right] - \frac{d_t}{\sqrt{v_t + \epsilon}}\right)$$

In terms of the decomposition in (48), we have $x_{t+1} = T_t(x_0, \ldots, x_t) + y_t$ where

$$T_t(x_0, \ldots, x_t) = (1 - \alpha_t\lambda)x_t - \alpha_t\mathbf{E}_t\left[\frac{d_t}{\sqrt{v_t + \epsilon}}\right]$$
$$y_t = \alpha_t\left(\mathbf{E}_t\left[\frac{d_t}{\sqrt{v_t + \epsilon}}\right] - \frac{d_t}{\sqrt{v_t + \epsilon}}\right).$$

Apparently we have $\mathbf{E}_t[y_t] = \mathbf{E}[y_t|x_0, \ldots, x_t] = 0$. We also have $\sum_{t=0}^\infty \mathbf{E}[\|y_t\|^2] < \infty$ with a bounded assumption on $y_t$ due to the assumption $\sum_{t=0}^\infty \alpha_t^2 < \infty$.

633 The squared distance between $T_t(x_0, \ldots, x_t)$ and $x_*$ is

$$\left\| T_t(x_0, \ldots, x_t) - x_* \right\|^2 = \left\| (1 - \alpha_t \lambda) x_t - \alpha_t \mathbf{E}_t \left[ \frac{d_t}{\sqrt{v_t + \epsilon}} \right] - x_* \right\|^2$$

$$= (1 - \alpha_t \lambda)^2 \| x_t - x_* \|^2 - 2\alpha_t (1 - \alpha_t \lambda) \left\langle x_t - x_*, \ \mathbf{E}_t \left[ \frac{d_t}{\sqrt{v_t + \epsilon}} \right] + \lambda x_* \right\rangle$$

$$+ \alpha_t^2 \left\| \mathbf{E}_t \left[ \frac{d_t}{\sqrt{v_t + \epsilon}} \right] + \lambda x_* \right\|^2$$

634 From Lemma D.5 and the aiming condition (36), we have

$$\left\langle x_t - x_*, \ \mathbf{E}_t \left[ \frac{d_t}{\sqrt{v_t + \epsilon}} \right] + \lambda x_* \right\rangle \geq \left\langle x_t - x_*, \ \frac{\mathbf{E}_t[d_t]}{\sqrt{\mathbf{E}_t[d_t^2]}} + \lambda x_* \right\rangle - \left( c_t \| \sqrt{\rho_t} \| + \mathcal{O}(\epsilon) + \mathcal{O}(\mathrm{Var}_t(v_t)) \right) \| x_t - x_* \|$$

$$\geq - \left( c_t \| \sqrt{\rho_t} \| + \mathcal{O}(\epsilon) + \mathcal{O}(\mathrm{Var}_t(v_t)) \right) \| x_t - x_* \|.$$

In addition, by the bounded assumption on $d_t$, there exist a constant $B$ such that

$$\left\| \mathbf{E}_t \left[ \frac{d_t}{\sqrt{v_t + \epsilon}} \right] + \lambda x_* \right\|^2 \leq B, \qquad \forall\, t \geq 0.$$

635 Together with $0 < 1 - \alpha_t \lambda < 1$, we conclude that

$$\left\| T_t(x_0, \ldots, x_t) - x_* \right\|^2 \leq (1 - \alpha_t \lambda)^2 \| x_t - x_* \|^2 + 2\alpha_t (1 - \alpha_t \lambda) \left( c_t \| \sqrt{\rho_t} \| + \mathcal{O}(\epsilon) + \mathcal{O}(\mathrm{Var}_t(v_t)) \right) \| x_t - x_* \| + \alpha_t^2 B$$

$$\leq (1 - \alpha_t \lambda)^2 \| x_t - x_* \|^2 + 2\alpha_t \left( c_t \| \sqrt{\rho_t} \| + \mathcal{O}(\epsilon) + \mathcal{O}(\mathrm{Var}_t(v_t)) \right) \| x_t - x_* \| + \alpha_t^2 B$$

$$= (1 + \alpha_t^2 \lambda^2) \| x_t - x_* \|^2 + \alpha_t \left( 2c \| \sqrt{\rho_t} \| + \mathcal{O}(\epsilon) + \mathcal{O}(\mathrm{Var}_t(v_t)) \right) \| x_t - x_* \|$$

$$- 2\alpha_t \lambda \| x_t - x_* \|^2 + \alpha_t^2 B$$

$$= (1 + \alpha_t^2 \lambda^2) \| x_t - x_* \|^2 + \alpha_t \left( \left( 2c \| \sqrt{\rho_t} \| + \mathcal{O}(\epsilon) + \mathcal{O}(\mathrm{Var}_t(v_t)) \right) \| x_t - x_* \| - \lambda \| x_t - x_* \|^2 \right)$$

$$- \alpha_t \lambda \| x_t - x_* \|^2 + \alpha_t^2 B.$$

636 We observe that there exist $\delta > 0$ such that

$$\| x_t - x_* \| \geq \delta \quad \Longrightarrow \quad \left( 2c \| \sqrt{\rho_t} \| + \mathcal{O}(\epsilon) + \mathcal{O}(\mathrm{Var}_t(v_t)) \right) \| x_t - x_* \| - \lambda \| x_t - x_* \|^2 \leq 0.$$

637 Therefore, $\| x_t - x_* \| \geq \delta$ implies

$$\left\| T_t(x_0, \ldots, x_t) - x_* \right\|^2 \leq (1 + \alpha_t^2 \lambda^2) \| x_t - x_* \|^2 - \alpha_t \lambda \| x_t - x_* \|^2 + \alpha_t^2 B.$$

638 Otherwise, when $\| x_t - x_* \| \leq \delta$, we have

$$\left\| T_t(x_0, \ldots, x_t) - x_* \right\|^2 \leq (1 - \alpha_t \lambda)^2 \delta^2 + 2\alpha_t \left( c \| \sqrt{\rho_t} \| \delta + \mathcal{O}(\epsilon) \right) + \alpha_t^2 B.$$

639 By defining $a_t$ as the right-hand side of the above inequality, i.e.,

$$a_t = (1 - \alpha_t \lambda)^2 \delta^2 + 2\alpha_t \left( c \| \sqrt{\rho_t} \| \delta + \mathcal{O}(\epsilon) \right) + \alpha_t^2 B, \tag{52}$$

640 we can combine the above two cases as

$$\left\| T_t(x_0, \ldots, x_t) - x_* \right\|^2 \leq \max \left\{ a_t, \ (1 + \alpha_t^2 \lambda^2) \| x_t - x_* \|^2 - \alpha_t \lambda \| x_t - x_* \|^2 + \alpha_t^2 B \right\}.$$

641 With the additional definition of

$$b_t = \alpha_t^2 \lambda^2, \qquad c_t = \alpha_t \lambda \| x_t - x_* \|^2, \qquad d_t = \alpha_t^2 B,$$

642 we arrive at the key inequality (49).

643 We are left to check the conditions in (50). Using the assumptions on $\{\alpha_t\}$, the definition in (52)
644 implies that $a_t$ converges and $\lim_{t \to \infty} a_t = \delta^2$. The conditions on $\{b_t\}$ and $\{d_t\}$ are automatically
645 satisfied. For $\{c_t\}$, if $\sum_{t=0}^{\infty} c_t < \infty$, then we must have $\| x_t - x_* \|^2 \to 0$ almost surely and the
646 conclusion of theorem holds trivially. Otherwise, $\sum_{t=0}^{\infty} c_t = \infty$ allows all the conditions in (50) to
647 hold, so we can invole Theorem D.1 to conclude the proof. $\qquad\square$

 # E    Biases and variances of second-moment estimators

649    Lemma 4.2 provides guidelines for choosing the second-moment estimator $v_t$, which should exhibit:

650    • low bias (i.e., low $\tau$);

651    • high signal-to-noise ratio (i.e., high SNR);

652    However, there is always a bias-variance tradeoff for various estimators $v_t$. Here are some examples:

653    1. **Sign-SGD** is equivalent to take $v_t = d_t^2$, exhibiting low bias and high variance

$$\text{Bias} = |\mathbf{E}_t[v_t] - \mathbf{E}_t[d_t^2]| = 0, \qquad \text{Variance} = \text{Var}_t(v_t) = \text{Var}_t(d_t),$$

654    with resulting update rule to be sign-SGD (with and without momentum corresponding to
655    $d_t = g_t$ and $d_t = m_t$ respectively):

$$x_{t+1} = x_t - \alpha_t \frac{d_t}{\sqrt{d_t^2 + \epsilon}} \approx x_t - \alpha_t \text{sign}(d_t).$$

656    2. **Standard SGD** is equivalent to take $v_t = c$ for some positive constant $c$, exhibiting high bias
657    and low variance

$$\text{Bias} = |\mathbf{E}_t[v_t] - \mathbf{E}_t[d_t^2]| = |c - \mathbf{E}_t[d_t^2]|, \qquad \text{Variance} = \text{Var}_t(v_t) = 0,$$

658    with resulting update rule to be SGD (with and without momentum corresponding to $d_t = g_t$
659    and $d_t = m_t$ respectively):

$$x_{t+1} = x_t - \alpha_t \frac{d_t}{\sqrt{c + \epsilon}} = x_t - \alpha_t' d_t,$$

660    where $\alpha_t' := \frac{\alpha_t}{\sqrt{c+\epsilon}}$.

661    3. **BCOS-m** uses $v_t = \text{EMA}_\beta(d_t^2)$, exhibiting non-trivial bias and low variance properties:

$$\begin{aligned}
\text{Bias} &= |\mathbf{E}_t[v_t] - \mathbf{E}_t[d_t^2]| \\
&= \left| \mathbf{E}_t\left[ \sum_{k=1}^{t}(1-\beta)\beta^{t-k}d_k^2 \right] - \mathbf{E}_t[d_t^2] \right| \\
&= \left| \sum_{k=1}^{t-1}(1-\beta)\beta^{t-k}d_k^2 + (1-\beta)\mathbf{E}_t\left[d_t^2\right] - \mathbf{E}_t[d_t^2] \right| \\
&= \left| \sum_{k=1}^{t-1}(1-\beta)\beta^{t-k}d_k^2 - \beta\mathbf{E}_t\left[d_t^2\right] \right|.
\end{aligned}$$

662    As for the variance, we get

$$\begin{aligned}
\text{Var}_t(v_t) &= \mathbf{E}_t\left[ \left( \sum_{k=1}^{t}(1-\beta)\beta^{t-k}d_k^2 - \sum_{k=1}^{t}(1-\beta)\beta^{t-k}\mathbf{E}_t[d_k^2] \right)^2 \right] \\
&= \mathbf{E}_t\left[ \left( \sum_{k=1}^{t-1}(1-\beta)\beta^{t-k}d_k^2 + (1-\beta)d_t^2 - \sum_{k=1}^{t-1}(1-\beta)\beta^{t-k}d_k^2 - (1-\beta)\mathbf{E}_t[d_k^2] \right)^2 \right] \\
&= (1-\beta)^2 \mathbf{E}_t\left[ \left( d_t^2 - \mathbf{E}_t[d_t^2] \right)^2 \right] \\
&= (1-\beta)^2 \text{Var}_t\left(d_t^2\right).
\end{aligned}$$

663    4. **Adam** uses estimator $v_t = \text{EMA}_{\beta_2}(g_t^2)$ with search direction $d_t = \text{EMA}_{\beta_1}(g_t)$: exhibiting
664    non-trivial bias and low variance properties:

$$\text{Bias} = |\mathbf{E}_t[v_t] - \mathbf{E}_t[d_t^2]|$$

$$= \left| \mathbf{E}_t \left[ \sum_{k=1}^{t} (1 - \beta_2) \beta_2^{t-k} g_k^2 \right] - \mathbf{E}_t \left[ \left( \sum_{k=1}^{t} (1 - \beta_1) \beta_1^{t-k} g_k \right)^2 \right] \right|$$

$$\leq \left| \sum_{k=1}^{t-1} (1 - \beta_2) \beta_2^{t-k} g_k^2 - \left( \sum_{k=1}^{t-1} (1 - \beta_1) \beta_1^{t-k} g_k \right)^2 \right|$$

$$+ 2 \left| \left( \sum_{k=1}^{t-1} (1 - \beta_1) \beta_1^{t-k} g_k \right) \mathbf{E}_t[g_t] \right| + \left| (1 - \beta_2) \mathbf{E}_t \left[ g_t^2 \right] - (1 - \beta_1)^2 \mathbf{E}_t[g_t^2] \right|.$$

As for the variance, we get

$$\text{Var}_t(v_t) = \mathbf{E}_t \left[ \left( \sum_{k=1}^{t} (1 - \beta_2) \beta_2^{t-k} g_k^2 - \sum_{k=1}^{t} (1 - \beta_2) \beta_2^{t-k} \mathbf{E}_t \left[ g_k^2 \right] \right)^2 \right]$$

$$= \mathbf{E}_t \left[ \left( \sum_{k=1}^{t-1} (1 - \beta_2) \beta_2^{t-k} g_k^2 + (1 - \beta_2) g_t^2 - \sum_{k=1}^{t-1} (1 - \beta_2) \beta_2^{t-k} g_k^2 - (1 - \beta_2) \mathbf{E}_t[g_t^2] \right)^2 \right]$$

$$= (1 - \beta_2)^2 \mathbf{E}_t \left[ \left( d_t^2 - \mathbf{E}_t[d_t^2] \right)^2 \right]$$

$$= (1 - \beta_2)^2 \text{Var}_t \left( d_t^2 \right).$$

5. **BCOS-c** uses estimator $v_t = (1 - (1 - \beta)^2) m_{t-1}^2 + (1 - \beta)^2 g_t^2$ with search direction $d_t = m_t = \text{EMA}_\beta(g_t) = \beta m_{t-1} + (1 - \beta) g_t$: exhibiting low bias and low variance properties:

$$\text{Bias} = |\mathbf{E}_t[v_t] - \mathbf{E}_t[d_t^2]|$$

$$= \left| \mathbf{E}_t \left[ (1 - (1 - \beta)^2) m_{t-1}^2 + (1 - \beta)^2 g_t^2 \right] - \mathbf{E}_t \left[ (\beta m_{t-1} + (1 - \beta) g_t)^2 \right] \right|$$

$$= \left| (2\beta - \beta^2) m_{t-1}^2 + (1 - \beta)^2 \mathbf{E}_t \left[ g_t^2 \right] - \beta^2 m_{t-1}^2 - 2\beta(1 - \beta) m_{t-1} \mathbf{E}_t \left[ g_t \right] - (1 - \beta) \mathbf{E}_t \left[ g_t^2 \right] \right|$$

$$= \left| (2\beta - 2\beta^2) m_{t-1}^2 - 2\beta(1 - \beta) m_{t-1} \mathbf{E}_t \left[ g_t \right] \right|$$

$$= 2\beta(1 - \beta) \left| m_{t-1} \left( m_{t-1} - \mathbf{E}_t \left[ g_t \right] \right) \right|$$

$$\text{Var}_t(v_t) = \mathbf{E}_t \left[ \left( (1 - (1 - \beta)^2) m_{t-1}^2 + (1 - \beta)^2 g_t^2 - (1 - (1 - \beta)^2) m_{t-1}^2 - (1 - \beta)^2 \mathbf{E}_t[g_t^2] \right)^2 \right]$$

$$= (1 - \beta)^4 \mathbf{E}_t \left[ \left( g_t^2 - \mathbf{E}_t[g_t^2] \right)^2 \right]$$

$$= (1 - \beta)^4 \text{Var}_t \left( g_t^2 \right).$$

