# OpenReview forum: "BCOS: A Method for Stochastic Approximation"
_NeurIPS.cc/2025/Conference — Submitted to NeurIPS 2025_

### Official Review · Reviewer_X2fe · 2025-07-02

**Clarity:** 4
**Significance:** 3
**Originality:** 3
**Rating:** 4
**Confidence:** 4

**Summary:**

This work proposes block-coordinate stepsizes and adaptive rules, which use online estimates of the second moment along each block coordinate. Adam can be understood as a special case of such heuristics. In conjunction with a simple conditional estimator, some optimizers are proposed. Convergence analysis is also provided.

**Questions:**

- Why are both $d_t$ and $g_t$ stochastic gradients above Eq (4)?
- It's mentioned after Eq (11) that for $\alpha_t$, consine decay or linear decay can be used. Instead, can BCOS be combined with recent schedule-free methods [1]?

[1] A. Defazio et al., The Road Less Scheduled

**Ethical Concerns:**

["NO or VERY MINOR ethics concerns only"]

**Final Justification:**

In my humble opinion, the flexibility and clean interpretation that this work provides is beneficial to the community. The aiming condition is indeed non-standard in modern ML optimization community. Yet, the authors are quite comprehensive and honest about its comparison to convexity (e.g., Appendix B).

I think it's true that more numerical experiments will strengthen the work, but: (i) I do not think the numerics are the main stength/point of this work, and (ii) I do think the existing experimental results exhibit the effect of stabilizing training in comparison to AdamW.

Therefore, I recommend acceptance.

**Limitations:**

Yes

**Paper Formatting Concerns:**

No major formatting issue

**Quality:**

3

**Strengths And Weaknesses:**

**Strength:**
- BCOS is a general framework from which popular algorithms like RMSprop and Adam can be interpreted as special cases. The derivation of optimal stepsizes, and the subsequent approximations/relaxations are well-motivated.
- Empirical evaluations seem reasonable, although the paper would be much more strengthened if evaluations are more extensive. Nevertheless, BCOS seems to stabilize

**Weakness:**
- Empirical evaluation: while the proposed method seems to stabilize the loss spikes that Adam/AdamW exhibits, with the current numerics, it is hard to conclude. For example, in Figure 4 right, it seems that BCOSW is slower than Adam, even though eventually it achieves better test accuracy. Neverthless, I think the flexibility and clean interpretation that this work provides is beneficial to the community, and perhaps better optimization methods can be designed inspired by this work.

---

> ### Author Rebuttal · Authors · 2025-07-31
>
> ## Answering reviewer's questions
>
> 1. Above Equation (4), the first $d_t=\nabla f(x,\xi)$ shows an example that $d_t$ can be assigned to be the stochastic gradient. Then we show another example that $d_t$ can be the momentum as in Equation (4), in this case, we let $g_t=\nabla f(x,\xi)$ and define $d_t$ as the EMA of $g_t$.
>
> 2. Yes, BCOS can be combined with the schedule-free method. It can be combined with any learning rate schedule methods that works with other optimizers. In this paper we mainly focus on the derivation, analysis and comparison with Adam/AdamW, so we use the most popular learning rate schedule of linear warmup followed by cosine decay.
>
> ## Clarification of main contributions
> We appreciate the reviewer's feedback on our strength and weakness, and would like to emphasize our main contributions outlined below:
>
> 1. We derive the BCOS family of stochastic approximation algorithms based on the fundamental principle of minimizing the expected distance from the next iterate to an (unknown) optimal point. This gives a novel interpretation of popular algorithms including RMSProp and Adam/AdamW. In particular, in contrast to common efforts that treat them as sophisticated optimization techniques such as diagonal preconditioning, we present the simple perspective of coordinate-wise contraction and constructing statistical estimators for the second moment of the search direction.
>
> 2. Within the framework of BCOS, we leverage the structure of stochastic momentum to derive a *conditional* estimator of the second moment. This leads to an algorithm that achieves competitive performance as Adam/AdamW but requires half of the optimizer states and fewer hyperparameters to tune. This alone makes a novel and strong contribution over the state-of-the-art AdamW optimizer.
>
> 3. We provide convergence analysis of the BCOS family based on a simple aiming condition that does not assume convexity or smoothness. While this condition may be unfamiliar to many researchers, it is an natural extension of the original assumption at the birth of stochastic approximation (see Answers to other reviews's questions). We obtain strong almost-sure convergence results and derive $O(1/t)$ convergence rate for the conceptual algorithm (using exact conditional second-moment).
>
> 4. We develop a new two-stage analysis technique for analyzing stochastic approximation algorithms. The first stage uses the aiming condition to prove almost sure convergence of a conceptual algorithm that assumes exact second-moment. The second stage extends the analysis to practical algorithms that use various online estimators for the second moment. We show that their performance guarantees depend on the bias-variance trade-offs of the estimators. This enables a common analysis for a broad family of algorithms, including RMSProp, sign-gradient/momentum, and Adam/AdamW.
>
> While our numerical experiments are limited in scope, they cover basic modalities (vision and language) and model architectures (ResNet, ViT, GPT), and demonstrate the key characteristics and potentials of the BCOS algorithms. In particular, the conditional BCOSW method enjoys smooth training/test loss curves for training transformer-based language models while AdamW often exhibits undesirable spikes.

---

> > ### Comment · Reviewer_X2fe · 2025-08-05
> >
> > Thank you for the detailed response and further clarifications. I followed the reviews by other reviewers, and it seems that there are many clarifications to be made, especially when $\lambda = 0$. Still, I believe the ML community can benefit from the clean exposition that this work provides, so I keep my original score.

---

> > > ### Author Response · Authors · 2025-08-06
> > >
> > > Thank you for your support! We agree that we need to add many clarifications in revising the paper, especially because our approach and main theoretical tools are quite different from the current machine learning optimization literature.
> > >
> > > We do have a full set of results for the case of $\lambda=0$, but choose not to include in the paper due to the page limite; instead, we focused on the $\lambda>0$ case which is more common in practice.

---

### Official Review · Reviewer_XLQ3 · 2025-07-03

**Clarity:** 2
**Significance:** 3
**Originality:** 2
**Rating:** 4
**Confidence:** 3

**Summary:**

This paper studies stochastic approximation with block-coordinate stepsizes. The authors propose  a new class of adaptive learning rate schemes yielding comparative performance to other adaptive step size choices such as ADAM. Analysis on the convergence of this new class of algorithms is provided and the theoretical findings of the paper are illustrated in simple numerical toy problems.

**Questions:**

-what assumptions on the noise \xi are necessary to ensure EMA is successful in estimating these conditional expectations (e.g. E_t [g_{t+1}])? Assumption B is related to that, but I would like to know more about the assumptions on \xi and when they are realistic. For example, the assumption that || g_t|| is bounded almost surely in theorem 4.3 is strong and needed for the practical implementation of the algorithm.

- x^* was never defined in the text. Is a stationary point of the gradient? Is it the argmin of F? How can you ensure it exists or is unique? you surely need Smoothness or some sort of convexity for that, no?

- Can the author provide realistic examples where their Assumption A would be satisfied? The quadratic objective example in Appendix B is not  very encouraging. How can this assumption be satisfied in reality?

- What is the loss function used in the experiments of the paper?

**Ethical Concerns:**

["NO or VERY MINOR ethics concerns only"]

**Final Justification:**

While my concerns about style remain, they may be less critical given that other reviewers were satisfied with the exposition and motivation of the results. I find the paper’s contributions both interesting and impactful for the ML community.

I disagree with reviewer SALE’s claim that asymptotic convergence to a small neighborhood of the solution necessarily weakens the paper’s contributions, but I am also not super familiar with other ADAM convergence papers as pointed out by reviewer 6Z8J . That said, I share SALE’s concern that the aiming condition should ideally be algorithm-agnostic. The authors explained in their rebuttal that this holds when d_t is a stochastic gradient measurement, but that they included a trajectory-dependent one to address more general settings (such as momentum algorithms). Since I am not familiar with momentum-based algorithms, I found the authors’ justification convincing.

The authors have addressed most of my original concerns, which has led me to increase my score. I remain unconvinced by the “realistic” example they provided for the aiming condition, but it appears that reviewer X2fe is satisfied with the discussion the authors included in the appendix.

Overall, I recommend acceptance.

**Limitations:**

I believe there are no potential negative societal impact on this work. The authors have stated the assumptions imposed in their paper.

**Paper Formatting Concerns:**

I have no concerns

**Quality:**

2

**Strengths And Weaknesses:**

This paper has interesting ideas and contributions with potential impact to the ML community. However, I believe that the current organization of the paper makes it hard to appreciate these contributions. There are a large number of symbols and very similar acronyms used early on in the main text that makes it hard to follow overall.

Also, the style of writing is a bit non-standard, mixing contributions and framework/methodology of analysis early in the main text, which can demotivates readers. In my opinion, the authors should present their algorithms first and later motivate them once the reader is  comfortable with the setup. The current organization takes several sidetracks along the way to expose the many variants of BCOS studied and makes it hard for the reader to be engaged. I also encourage the readers to proofread their paper as I have found several typos and confusing passages.

---

> ### Author Rebuttal · Authors · 2025-07-31
>
> We first address a few common questions raised by the reviewers, then respond to the specific questions by this reviewer.
>
> # Addressing several common questions
> ## 1. Aiming condition
> Aiming condition appeared in the original Robbins-Monro paper [38] on stochastic approximation as the main assumption to guarantee convergence (single dimension case). The vector version $\langle x-x_*,\mathbf{E}[g(x,\xi)]\rangle>0$ appeared in the multi-dimensional extension by Blum [3]. Both classical works require slightly stronger aiming conditions to prove almost sure convergence, and a typical sufficient condition is
> $$L\|x-x_*\|^2 \geq \langle x-x_*, \mathbf{E}[g(x,\xi)]\rangle \geq \mu\|x-x_*\|^2$$
> for some $L\geq\mu>0$. Indeed, most results in our current understanding of stochastic gradient methods can be derived as special cases of the stochastic approximation literature developed in 1950's. If $g(x,\xi)$ is the stochastic gradient of some loss function, the above aiming condition is much more general than (strong) convexity and smoothness.
>
> **Example.**
> For simplicity, consider the 1-dimensional case $x\in\mathbb{R}$ and without loss of generality let $x_*=0$. Let $g(x):=\mathbf{E}[g(x,\xi)]$ be any continuous function that satisfies $g(x) > 0$ for $x> 0$ and $g(x)<0$ for $x<0$, which leads to
> $$
> \langle x-x_*,g(x)\rangle =x \cdot g(x)>0, \qquad \forall x\neq 0.
> $$
> Suppose $g(x)$ is the gradient of some loss function $f(x)$. Then this condition is much weaker than convexity, which would require monotonicity of $g(x)$, that is, $(x-y)(g(x)-g(y))>0$ for all $x,y\in\mathbb{R}$.
> For example, consider
> $$g(x)=x(x-3)^2(x+2)^2+\lambda x,$$
> which is a non-monotone polynomial, thus it is the gradient of a nonconvex polynomial. But this function satisfies the aiming condition for $x_*=0$ and for any $\lambda\geq 0$. Apparently, such examples abound.
>
> **Our aiming condition (Assumption A).**
> When the search direction $d_t$ is independent of the past trajectory (e.g., stochastic gradient with i.i.d. noise), our aiming condition in Equation (23) is equivalent to
> $$
> \left\langle x-x_*, \frac{\mathbf{E}[d(x,\xi)]}{\sqrt{\mathbf{E}[d(x,\xi)^2]}} + \lambda x \right\rangle \geq \lambda\|x-x_*\|^2,
> $$
> where we used the definition of $\rho_t$ to expand
> $\sqrt{\rho_t}\odot\textup{sign}(\mathbf{E}_t[d_t])=\frac{\mathbf{E}_t[d_t]}{\sqrt{\mathbf{E}_t[d^2]}}=\frac{\mathbf{E}[d(x,\xi)]}{\sqrt{\mathbf{E}[d(x,\xi)^2]}}$.
>    - Our aiming condition is an extension of the classical one by incorporating coordinate-wise scaling, which is a natural adaptation to handle modern optimizers with diagonal scaling.
>    - We do not need upper bound on the inner product as in the classical aiming condition (which is implied by smoothness), due to the coordinate-wise normalization by $\sqrt{\mathbf{E}_t[d_t^2]}$.
>    - Diagonal scaling in the aiming condition causes subtle changes on the problem structure. For example, the corresponding optimal point $x_*$ is now different. This is similar to the fact that adding the regularization $\lambda\|x\|^2$ to the loss function changes the optimal point of the overall loss. Moverover,
>        * It no longer contains all convex functions as special cases, as shown by our counter example in Appendix B. We use this very ill-conditioned quadratic function to illustrate the point. The new aiming condition still holds for most convex functions that are not severely ill-conditioning (depending on the scaling). Meanwhile, diagonal scaling also opens up additional non-convex functions to satisfy the aiming condition. Therefore, it is overlapping with convexity.
>        * Since the optimal solution $x_*$ is changed with diagonal scaling, our example of ill-conditioned quadratic may still satisfy the aiming condition with respect to a different $x_*$.
>        * The question of finding the optimal solution $x_*$ given scaled directions $d(x)$ falls into the standard form of *variational inequality* $\langle x-x_*,d(x)\rangle\geq 0$ or the strengthened version $\langle x-x_*, d(x)+\lambda x\rangle\geq\lambda\|x\|^2$. There is a vast literature on solving variational inequalities and its connection to nonlinear programming. BCOS is a class of effective methods for solving *stochastic variational inequalities*.
>    - To handle the general case of correlated search directions, such as using stochastic momentum, we strengthen the aiming condition to be trajectory-dependent, as stated in Assumption A. Admittedly, such a condition is difficult to check a priori. But we have computational evidence that they mostly hold in our experiments (by recording the final weights of a trained model as $x_*$ and repeat the experiments with same random seeds in initialization and data sampling.)
>
> ## 2. Convergence analysis when $\lambda=0$
> We gave some brief comments on convergence analysis when $\lambda=0$ at the bottom of page 7. To further elaborate:
>   - When $\lambda=0$, we need slightly stronger aiming condition to provide similar convergence guarantees. For example, it is sufficient to have
>     $$
>     \left\langle x-x_*, \frac{\mathbf{E}[d(x,\xi)]}{\sqrt{\mathbf{E}[d(x,\xi)^2]}} \right\rangle \geq \mu \|x-x_*\|^2,
>     $$
>     for some $\mu>0$. This can be the case of assimilating $\lambda x$ into the search direction $d(x,\xi)$, where we do not get the full strength of regularization $\lambda$ on the right hand side (due to diagonal scaling), but only settle for some $\mu<\lambda$.
>   - For convergence rate, with any $\mu>0$ in the above condition, we can still obtain $O(1/t)$ rate for the conceptual algorithm. Moverover, we can remove the condition $\alpha\lambda>1/2$ and settle with $O(1/t^p)$ where $p<1$ can be arbitrarily close to 1. These are all direct consequences of classical stochastic approximation [6].
>
> ## 3. Bias and variance bounds for the second-moment estimator
> Our analysis of the practical BCOS algorithms relies on Assumption B. We can always construct an estimator with $\tau<1$ with sufficiently large $\epsilon$.
>    - We listed the bias-variance trade-offs of several popular algorithms at the bottom of page 8. Apparently, more concrete bounds can be derived depending on $\tau$ and $\epsilon$, as well as the smoothness constants of the loss function. But their expressions can be quite tedious and loose. We will try to include some refined bounds in the revision, but they do not dictate our fundamental contributions.
>    - The exact magnitudes of the bias and variance of the estimators do not affect stability of the algorithms. The stability or convergence of the algorithms are determined by the aiming condition for the conceptual algorithm. When the conceptual algorithm is stable and converge, the actual bias and variance of the practical algorithms only affect the quality of solution (radius of neighborhood of $x_*$).
>
> # Answering specific questions of the reviewer
>
> 1. We do not need specific conditions on $\xi$. We only need to assume $\mathbf{E}[d_t^2]$ is bounded, and there always exists an estimator satisfying Assumption B with $\tau<1$ with sufficiently large $\epsilon$. We believe that the assumption of $d_t$ being bounded (the bound can be arbitrarily large) is reasonable both in theory and practice. In fact, we can even avoid assuming $d_t$ is bounded by assuming $d_t^2$ has bounded variance, which we will add in the revision.
>
> 2. We assume there exist an $x_*$ such that the aiming condition holds. In the full block case we recover the classical Aiming condition as discussed above, which is the minimizer of the regularized loss function whose corresponding gradient is $d_t$. If $d_t$ is the momentum, $x_*$ may not be the minimizer of the regularized loss function. In general, it is the solution of the variational inequality defined by Equation (23), which has a more explicit form in our explanation of the Aiming condition above (addressing common questions section).
>
> 3. Realistic examples satisfying Assumption A is given above in the discussion on Aiming Condition. Specifically, $g(x)=x(x-3)^2(x+2)^2+\lambda x$ for any $\lambda>0$. Apparently, such examples abound in polynomials.
>
> 4. The loss function we used in the experiments is the standard cross-entropy for both image classification and language modeling.
>
> # Main contributions
>
> Finally, we remind the reviewer of our main contributions:
> 1. We derive BCOS family from simple principle of coordinate-wise contraction, which gives Adam(W) a novel interpretation.
> 2. The BCOS framework enables us to develop a new variant by leveraging a simple conditional estimator, which obtains competitive performance against AdamW but only use half of its optimizer states and less hyper parameter to tune. This alone makes a novel and strong contribution over the state-of-the-art AdamW optimizer.
> 3. We provide convergence of a broad family of algorithms using an *Aiming Condition*, which does not assume convexity or smoothness. While this condition may be unfamiliar to many researchers, it is a natural extension of the original assumption at the birth of stochastic approximation. We reintroduce the rich literature on stochastic approximation developed in the 1950's to the contemporary research community.
> 4. We develop a new two-stage analysis technique for analyzing stochastic approximation algorithms. The first stage uses the aiming condition to prove almost sure convergence of a conceptual algorithm that assumes exact second-moment. The second stage extends the analysis to practical algorithms that use various online estimators for the second moment. We show that their performance guarantees depend on the bias-variance trade-offs of the estimators. This enables a common analysis for a broad family of algorithms, including RMSProp, sign-gradient/momentum, and Adam/AdamW.

---

> > ### Comment · Reviewer_XLQ3 · 2025-08-06
> >
> > I thank the authors for their detailed responses. My concerns have been addressed, and I will raise my score accordingly. That said, I believe the final version would benefit from further contextualization/explanation/motivation of the assumed aiming condition.

---

> > > ### Author Response · Authors · 2025-08-06
> > >
> > > Thank you for your support and feedback! We will provide more context and explanation for the aiming condition in the revision.

---

### Official Review · Reviewer_6Z8J · 2025-07-03

**Clarity:** 3
**Significance:** 2
**Originality:** 3
**Rating:** 3
**Confidence:** 3

**Summary:**

This paper studies the optimization problem and proposes a novel family of optimizers called BCOS. BCOS is derived from the block-coordinate step size for the updates. Theoretical analysis and empirical validation for the proposed algorithms are provided.

**Questions:**

Please refer to the weakness part.

**Ethical Concerns:**

["NO or VERY MINOR ethics concerns only"]

**Final Justification:**

In my opinion, I think the motivation and the overall derivation are interesting and should be acceptable for NeurIPS. But for now, I keep my rating 3 mainly because the issues remain after rebuttal: 1) As Reviewer SALE mentioned, the theoretical results themselves may not be strong enough, especially when compared with other Adam convergence papers. 2) I don't think the authors' explanation for their empirical validation setting is convincing enough. I still don't understand why they tune $ \beta_1, \beta_2 $ with pretuned learning rates. I think a clearer empirical justification is needed.

The theoretical justification of the aiming condition provided in the rebuttal period should be good for improving the paper, and I think experimental validation, e.g., verifying the condition in some simple neural networks, will also be welcome. Also, I appreciate the authors' answers to my question about AdaGrad, but I don't think they got the point, though this is not a major point for my evaluation.

**Limitations:**

Yes.

**Paper Formatting Concerns:**

No.

**Quality:**

3

**Strengths And Weaknesses:**

**Strengths**:
1. The paper is well-written and easy to follow. The derivation and motivation of BCOS are clearly explained.

2. The derivation of BCOS is interesting, which not only gives birth to BCOS, but also provides some intuitions into Adam and other adaptive gradient methods.

**Weakness**:
1. In the derivation, we focus on minimizing the 2-norm of $x - x^*$, while this may not be a natural metric for analyzing optimizers with coordinate-wise step sizes. In fact, mostly we consider a different norm with respect to the preconditioner $||\cdot||_{\Lambda}$ and obtain convergence with respect to a different norm (e.g., $||\cdot||_\infty$ for AdaGrad as analyzed by the original paper by Duchi et. al.). What will happen when we consider some alternative metrics (different norms)? Will the derivation still hold, or will we get something else?

2. It is still weird to use Assumption A for a rigorous convergence proof, since it is definitely uncommon and really depends on the trajectory. I think that at least some examples for the functions satisfying Assumption A and/or some empirical validation of the assumption are required. When $d_t$ is momentum as set in the algorithms, it doesn't seem to satisfy the independence condition.

3. The experiments are done with fixed step sizes and only tune other less important parameters $\beta_1, \beta_2$, which is kind of strange. At least figure 1 may need more refinement, since the spike can possibly be fixed by a well-tuned step size for AdamW.

---

> ### Author Rebuttal · Authors · 2025-07-31
>
> We first address a few common questions raised by the reviewers, then respond to the specific questions by this reviewer.
>
> # Addressing several common questions
> ## 1. Aiming condition
> Aiming condition appeared in the original Robbins-Monro paper [38] on stochastic approximation as the main assumption to guarantee convergence (single dimension case). The vector version $\langle x-x_*,\mathbf{E}[g(x,\xi)]\rangle>0$ appeared in the multi-dimensional extension by Blum [3]. Both classical works require slightly stronger aiming conditions to prove almost sure convergence, and a typical sufficient condition is
> $$L\|x-x_*\|^2 \geq \langle x-x_*, \mathbf{E}[g(x,\xi)]\rangle \geq \mu\|x-x_*\|^2$$
> for some $L\geq\mu>0$. Indeed, most results in our current understanding of stochastic gradient methods can be derived as special cases of the stochastic approximation literature developed in 1950's. If $g(x,\xi)$ is the stochastic gradient of some loss function, the above aiming condition is much more general than (strong) convexity and smoothness.
>
> **Example.**
> For simplicity, consider the 1-dimensional case $x\in\mathbb{R}$ and without loss of generality let $x_*=0$. Let $g(x):=\mathbf{E}[g(x,\xi)]$ be any continuous function that satisfies $g(x) > 0$ for $x> 0$ and $g(x)<0$ for $x<0$, which leads to
> $$
> \langle x-x_*,g(x)\rangle =x \cdot g(x)>0, \qquad \forall x\neq 0.
> $$
> Suppose $g(x)$ is the gradient of some loss function $f(x)$. Then this condition is much weaker than convexity, which would require monotonicity of $g(x)$, that is, $(x-y)(g(x)-g(y))>0$ for all $x,y\in\mathbb{R}$.
> For example, consider
> $$g(x)=x(x-3)^2(x+2)^2+\lambda x,$$
> which is a non-monotone polynomial, thus it is the gradient of a nonconvex polynomial. But this function satisfies the aiming condition for $x_*=0$ and for any $\lambda\geq 0$. Apparently, such examples abound.
>
> **Our aiming condition (Assumption A).**
> When the search direction $d_t$ is independent of the past trajectory (e.g., stochastic gradient with i.i.d. noise), our aiming condition in Equation (23) is equivalent to
> $$
> \left\langle x-x_*, \frac{\mathbf{E}[d(x,\xi)]}{\sqrt{\mathbf{E}[d(x,\xi)^2]}} + \lambda x \right\rangle \geq \lambda\|x-x_*\|^2,
> $$
> where we used the definition of $\rho_t$ to expand
> $\sqrt{\rho_t}\odot\textup{sign}(\mathbf{E}_t[d_t])=\frac{\mathbf{E}_t[d_t]}{\sqrt{\mathbf{E}_t[d^2]}}=\frac{\mathbf{E}[d(x,\xi)]}{\sqrt{\mathbf{E}[d(x,\xi)^2]}}$.
>    - Our aiming condition is an extension of the classical one by incorporating coordinate-wise scaling, which is a natural adaptation to handle modern optimizers with diagonal scaling.
>    - We do not need upper bound on the inner product as in the classical aiming condition (which is implied by smoothness), due to the coordinate-wise normalization by $\sqrt{\mathbf{E}_t[d_t^2]}$.
>    - Diagonal scaling in the aiming condition causes subtle changes on the problem structure. For example, the corresponding optimal point $x_*$ is now different. This is similar to the fact that adding the regularization $\lambda\|x\|^2$ to the loss function changes the optimal point of the overall loss. Moverover,
>        * It no longer contains all convex functions as special cases, as shown by our counter example in Appendix B. We use this very ill-conditioned quadratic function to illustrate the point. The new aiming condition still holds for most convex functions that are not severely ill-conditioning (depending on the scaling). Meanwhile, diagonal scaling also opens up additional non-convex functions to satisfy the aiming condition. Therefore, it is overlapping with convexity.
>        * Since the optimal solution $x_*$ is changed with diagonal scaling, our example of ill-conditioned quadratic may still satisfy the aiming condition with respect to a different $x_*$.
>        * The question of finding the optimal solution $x_*$ given scaled directions $d(x)$ falls into the standard form of *variational inequality* $\langle x-x_*,d(x)\rangle\geq 0$ or the strengthened version $\langle x-x_*, d(x)+\lambda x\rangle\geq\lambda\|x\|^2$. There is a vast literature on solving variational inequalities and its connection to nonlinear programming. BCOS is a class of effective methods for solving *stochastic variational inequalities*.
>    - To handle the general case of correlated search directions, such as using stochastic momentum, we strengthen the aiming condition to be trajectory-dependent, as stated in Assumption A. Admittedly, such a condition is difficult to check a priori. But we have computational evidence that they mostly hold in our experiments (by recording the final weights of a trained model as $x_*$ and repeat the experiments with same random seeds in initialization and data sampling.)
>
> ## 2. Convergence analysis when $\lambda=0$
> We gave some brief comments on convergence analysis when $\lambda=0$ at the bottom of page 7. To further elaborate:
>   - When $\lambda=0$, we need slightly stronger aiming condition to provide similar convergence guarantees. For example, it is sufficient to have
>     $$
>     \left\langle x-x_*, \frac{\mathbf{E}[d(x,\xi)]}{\sqrt{\mathbf{E}[d(x,\xi)^2]}} \right\rangle \geq \mu \|x-x_*\|^2,
>     $$
>     for some $\mu>0$. This can be the case of assimilating $\lambda x$ into the search direction $d(x,\xi)$, where we do not get the full strength of regularization $\lambda$ on the right hand side (due to diagonal scaling), but only settle for some $\mu<\lambda$.
>   - For convergence rate, with any $\mu>0$ in the above condition, we can still obtain $O(1/t)$ rate for the conceptual algorithm. Moverover, we can remove the condition $\alpha\lambda>1/2$ and settle with $O(1/t^p)$ where $p<1$ can be arbitrarily close to 1. These are all direct consequences of classical stochastic approximation [6].
>
> ## 3. Bias and variance bounds for the second-moment estimator
> Our analysis of the practical BCOS algorithms relies on Assumption B. We can always construct an estimator with $\tau<1$ with sufficiently large $\epsilon$.
>    - We listed the bias-variance trade-offs of several popular algorithms at the bottom of page 8. Apparently, more concrete bounds can be derived depending on $\tau$ and $\epsilon$, as well as the smoothness constants of the loss function. But their expressions can be quite tedious and loose. We will try to include some refined bounds in the revision, but they do not dictate our fundamental contributions.
>    - The exact magnitudes of the bias and variance of the estimators do not affect stability of the algorithms. The stability or convergence of the algorithms are determined by the aiming condition for the conceptual algorithm. When the conceptual algorithm is stable and converge, the actual bias and variance of the practical algorithms only affect the quality of solution (radius of neighborhood of $x_*$).
>
>
> # Answering specific questions of the reviewer
>
> 1. The squared 2-norm of $x_t-x_\ast$ is indeed the most nature metric for analysis based on distance to the target point $x_\ast$. Especially it can be decomposed into sums of coordinate-wise squared distances. This is very different from using different norms to shape the geometry of the descent direction (e.g., 2-norm for steepest descent and other norms leading to different directions). For our framework, squared 2-norm is the only one that makes sense.
>
> 2. Assumption A is indeed an extension of the aiming condition that dominated the stochastic approximation literature at it birth in the 1950's. It is a more general and powerful condition than modern assumptions restricted to convexity and/or smoothness. Please see the above general discussion on Aiming condition, as well as the examples given there. When $d_t=m_t$, the search directions are correlated across different iterations, so the general form of Assumption A is trajectory dependent. Admittedly it is difficult to give a priori analytic characterization, but we can add empirical evidences of the aiming condition as discussed above.
>
> 3.  We do not use fixed stepsize. As we explain in lines 250-252, we use linear LR warmup followed by cosine decay, with clearly stated number of steps, peak learning rates and cosine decay ratio. In the figure legends and captions, $\alpha$ means the peak learning rate.
> # Main contributions
>
> Finally, we remind the reviewer of our main contributions:
> 1. We derive BCOS family from simple principle of coordinate-wise contraction, which gives Adam(W) a novel interpretation.
> 2. The BCOS framework enables us to develop a new variant by leveraging a simple conditional estimator, which obtains competitive performance against AdamW but only use half of its optimizer states and less hyper parameter to tune. This alone makes a novel and strong contribution over the state-of-the-art AdamW optimizer.
> 3. We provide convergence of a broad family of algorithms using an *Aiming Condition*, which does not assume convexity or smoothness. While this condition may be unfamiliar to many researchers, it is a natural extension of the original assumption at the birth of stochastic approximation. We reintroduce the rich literature on stochastic approximation developed in the 1950's to the contemporary research community.
> 4. We develop a new two-stage analysis technique for analyzing stochastic approximation algorithms. The first stage uses the aiming condition to prove almost sure convergence of a conceptual algorithm that assumes exact second-moment. The second stage extends the analysis to practical algorithms that use various online estimators for the second moment. We show that their performance guarantees depend on the bias-variance trade-offs of the estimators. This enables a common analysis for a broad family of algorithms, including RMSProp, sign-gradient/momentum, and Adam/AdamW.

---

> > ### Comment · Reviewer_6Z8J · 2025-08-04
> > **Reply to the Rebuttal**
> >
> > Thanks for the reply. However, I don't think the provided context properly addressed my questions.
> > 1. Sorry for the unsuccessfully generated context. Let me restate what I mean for the first point here. Only considering the 2-norm for algorithm derivation may not be a good choice, since the convergence of AdaGrad (Duchi et al.) depends on some other measures like infinite-norm or some kind of Mahalanobis Norm. In fact, if we only consider the Euclidean distance for proving the convergence theorem, AdaGrad shows no better or even worse rate compared to SGD, failing to catch why we want to derive adaptive gradient methods. I suggest that the authors at least draw some attention to this.
> > 2. The assumption is not a common one in the optimization literature, and it is also not a generalization of any common assumption. I suggest at least adding more theoretical and/or justification, which is definitely not enough in the current version of paper.
> > 3. By a fixed step size, apparently I mean you choose a step size (schedule) without even tuning, while you tune the less important hyperparameters like $ \beta_1, \beta_2 $, thus I question the validity of the experiments.

---

> > > ### Author Response · Authors · 2025-08-05
> > >
> > > Here are our further clarifications to your questions.
> > >
> > >   (1) There are two fundamental paradigms for convergence analysis of optimization algorithms:
> > >   - One is to analyze the function descent property, where we find the steepest descent direction with respect to different norms measuring deviation from $x_t$.
> > >   - The other one is to analyze directly the reduction of distance from the optimal point during each step.
> > >
> > >   The first approach is more suitable for minimizing smooth functions, while the second one is more suitable for minimizing nonsmooth functions or with subgradient or stochastic gradient oracle. Fundamentally because subgradients or stochastic gradients are in general NOT descent directions. We believe the reviewer is talking about the first approach, while our paper uses the second approach. For the first approach, we understand that there are different norms one can choose to shape the descent direction. But for the second approach, 2-norm is the most convenient choice. We derive the coordinate-wise stepsize by decomposing the squared 2-norm into simple quadratics in each coordinates, which is not feasible for other norms.
> > >
> > >   (2) We agree with the reviewer that the Aiming condition is not a common one in contemporary optimization literature, and we need to add more explanation when revising the paper as we have done in the rebuttal on common questions.
> > >   However, we also view this as one of our main contributions: to break away from the mainstream assumptions of (strong) convexity and/or smoothness in analyzing stochastic gradient methods.
> > >   - As we explain in the rebuttal, the classical Aiming Condition (developed in 1950's, starting from the Robbins and Monro Paper [38]) is more general and powerful than the convexity and/or smoothness framework. Indeed, most our current understanding of stochastic gradient methods can be derived as special cases of these early developments.
> > >   - Our aiming condition extends the classical condition by incorporating (block) coordinate-wise step sizes. Another improvement over classical aiming condition is that we no longer need a quadratic upper bound on the inner product (see rebuttal on common questions \#1), due to the proper scaling of $d_t$ by $\sqrt{\mathbf{E_t}[d_t^2]}$. This is roughly equivalent to removing the smoothness condition, representing another major deviation of our theory from "standard" setting of convexity and/or smoothness.
> > >   - We hope this paper can raise awareness of the rich literature on stochastic approximation developed in the 1950's, which contain more general and powerful tools that are mostly ignored by the contemporary research community. They provide an alternative, actually more fundamental, understanding of state-of-the-art optimizers such as Adam/AdamW, and lead to novel variants such as BCOS with conditional estimators.
> > >
> > >   (3) For the language model task on on GPT2, we did tune the stepsize (schedule). We mentioned on Line 251 that they are ``chosen based on a coarse sweep'', meaning that we did sweep the stepsize (learning rate) in the range $\{0.0001, 0.0003, 0.001, 0.002, 0.004\}$, and choose the best peak learning rate $\alpha=0.002$. This learning rate worked best for tuning other hyperparameters in the limited range of $\beta_1$ and $\beta_2$. This is a common task and we found these hyperparameter choices align with the best practices shared by many open source projects.

---

> > > > ### Comment · Reviewer_6Z8J · 2025-08-06
> > > >
> > > > Thanks for the replies to my questions.
> > > >
> > > > 1. I basically don't get the point. I should note that the original AdaGrad paper did the analysis under **nonsmooth** settings, which should fall into the second approach as the authors mentioned. The convergence of AdaGrad would rely on $ || x - x^*||_\infty $, and the proof highly relies on the involvement of the Mahalanobis Norm $ || x ||_S^2 = x^\top S x $, but not the Euclidean norm. I highly recommend that the authors at least read the AdaGrad paper if they really misunderstood this. Also, based on the AdaGrad analysis, the Mahalanobis Norm should also be convenient in the analysis, since it shares similar properties as the Euclidean norm. I am not saying this is a necessary thing for the paper, but I suggest you at least get more understanding and try.
> > > >
> > > > 2. Thanks for the clarification. I think it would definitely improve the quality of the paper by involving more discussions on the assumption.
> > > >
> > > > 3. I appreciate the author's clarification on point 3, and suggest that the author include the detailed sweep grid in the paper, though I am still confused about how the author did the sweep. Is the learning rate chosen for achieving optimal performance of Adam? Or achieving optimal performance of BCOS? Also, since the target of the experiment is to compare BCOS with Adam, I am still confused about why you tuned the less important hyperparameters like $ \beta_1, \beta_2 $, while leaving the learning rate (which people usually tune the most) just the same throughout the experiments? This doesn't seem to be a reasonable comparison setting to me.
> > > >
> > > > To conclude, I think the motivation of the paper is basically interesting, with insights looking into the derivation of BCOS, and about the aiming condition. But some issues still remained. I would like to keep my score for now.

---

> > > > > ### Author Response · Authors · 2025-08-08
> > > > >
> > > > > 1. We are very familiar with the AdaGrad paper. The reviewer is correct that the AdaGrad paper considers the nonsmooth setting thus their analysis belongs to the second regime we explained above.
> > > > >
> > > > >     For the second regime, "2-norm is the most convenient choice," and the analysis is not hard to generalize to quadratic norms or Mahalanobis norms, because the squared distance under such norms can all be expanded as sum of weighted coordinate-wise squares and inner products. So we agree with the reviewer the possiblility of extension to "different norms" in this sense.
> > > > >
> > > > >     We are contrasting "2-norm" (or more general quadratic norms) with infinity norm because the squared distance measured by the infinity norm cannot be decomposed into sum of coordinate-wise entities that allow our analysis framework.
> > > > >
> > > > >     That being said, we can still have the infinity norm appearing in the bounds for convergence rates or regret bounds. This is exactly what the AdaGrad paper has. Notice that the quantity $\Vert x_t-x_* \Vert_\infty^2$ appeared in the AdaGrad paper in their Equation (14) on page 2131, where they used Hölder's inequality to bound an inner product of two vectors, one is $(x_t-x_*)^2$ (element-wise square of vector) and the other is $s_t$ whose coordinate $s_{t,i}$ is the norm of the concatenated gradients of coordinate $i$ from step 1 to $t$. The reason they use Hölder's inequality is because $\Vert s_t \Vert_1$ has a special form/meaning in their context, so $\Vert x_t-x_* \Vert_\infty^2$ appears in their regret bound.
> > > > >
> > > > >     For us, in a similar situation in our analysis, Equation (51) below Line 627, we used Cauchy-Schwartz to bound an inner product with the product of 2-norms. Indeed, we can also use Holder inequality to bound the inner product there by $\Vert x_t-x_* \Vert_1 \cdot \Vert c_t E_t[d_t]/\sqrt{E_t[d_t^2]} + \cdots\Vert_\infty$. However, this eventually does not lead to improved bound and that's why we did not go to this route --- it really depends on the structure of specific problems.
> > > > >
> > > > >     Getting back to the reviewer's original question, yes, we may conduct the analysis in a more general quadratic norm instead of the simple 2-norm. But this does not lead to any essential changes of our analysis, possibly with different constants on the bounds.
> > > > >
> > > > >   2. We will add more discussions of Aiming condition in revising the paper. We thought it should be well-known as the original aiming condition appeared in the original Robbins-Monro paper and is the foundation of almost every classical paper on stochastic approximation.
> > > > >
> > > > >   3. We did tune the learning rate as we explained above, "and choose the best peak learning rate $\alpha=0.002$. This learning rate worked best for other hyperparameters in the limited range of $\beta_1$ and $\beta_2$" we tried in the experiments.
> > > > >
> > > > >
> > > > > **We thank the reviewer for recognizing the interesting aspects of our paper: motivation, insights, aiming condition. But we remind the reviewer again that our main contributions are well beyond:**
> > > > >
> > > > > - The BCOS framework enables us to develop a new variant with a simple conditional estimator, which obtains competitive performance against AdamW but only use half of its optimizer states and less hyper parameter to tune. This alone makes a novel and strong contribution. It is one of the very few essential progresses on optimizers since Adam and AdamW.
> > > > > - We introduce a new Aiming Condition, which does not assume convexity or smoothness (more general and powerful) and breaks away from common practice. By doing so, we reintroduce the rich literature on stochastic approximation developed in the 1950's to the contemporary research community.
> > > > > - The BCOS framework gives Adam(W) a novel interpretation and we develop a two-stage analysis framework that applies to a broad family of algorithms, including RMSProp, sign-gradient/momentum, and Adam/AdamW (see Lines 236-244).
> > > > >
> > > > > These contributions are of fundamental nature and far from incremental. So it is not clear to us what are "some issues still remained"?
> > > > >
> > > > > Because of the new perspective and new analytical tools we introduce, it may take time for acceptance by the research community. We believe NeurIPS should be the perfect venue to promote new ideas and methodologies, and hope the reviewer would agree that our work passes the bar.

---

### Official Review · Reviewer_SALE · 2025-07-05

**Clarity:** 2
**Significance:** 2
**Originality:** 2
**Rating:** 3
**Confidence:** 4

**Summary:**

The paper studies a class of algorithms called BCOS (Block-Coordinate Step sizes) that resembles Adam in terms of first and second moment computations, but has less memeory overhead with fewer tunable parameters. The derivation is based on the convergence analysis with respect to the distance to the solution, and selecting the best possible step size yields a conceptual algorithm that is not implementable. The practically viable variants are obtained by applying a set of heuristic simplifications, which seem to be selected with the intention to obtain an update rule similar to Adam. The main differences are:

1.	the use of one $\beta$ variable instead of two,

2.	second moment computation has multiple interpretations beyond Adam’s,

3.	and the analysis is done for general update directions (which is possible by assumptions that control the bias of $v_t$ and impose an alignment condition on the preconditioned update direction with respect to the optimum of the objective)

The proposed variants are compared against Adam and AdamW for training language and vision models.

**Questions:**

1.	I see that the goal is to obtain a state storage-efficient method, therefore all the derivations are done to obtain such a method, but the intuition behind them are not strong otherwise. For instance, would the choice $E[g_t] = g_t$ and $E[g_t^2] = m_t^2$ (or a variant where the iteration count $t$ is shifted) work in a similar way? In terms of storage efficieny, comparing against AdaGrad might be as relevant.

2.	The convergence rate in Theorem 4.2 has dimension dependence and eliminating it would require $\alpha$ to be inversely proportional to $n$. What choices of $\alpha, \lambda$ is proposed? What happens when $\lambda = 0$?

3.	One might design a similar method by optimizing the choice of the step size before taking expectation. Would it be necessarily worse than your selection (after expectation)?

4.	Could you please explain in details how you obtain the result for Lemma 4.2 from the last inequality in line 602 in the Appendix?

5.	For the case of no weight decay (e.f., BCOS with $\lambda = 0$), what happens to the result of Theorem 4.3? Would it imply zero variance for $v_t$? What happens to the bias bounds?

6.	Could you please give an example of a non-convex function in the presence of Assumption 1 when $\lambda > 0$ and $d_t = m_t$?

7.	How could the bias bound be compared to more standard bounded variance? Could you obtain a similar bias bound by assuming bounded variance on $g_t$ and setting $d_t = g_t$ or $d_t = m_t$? Also, it is important to note that Theorem 4.3 assumes bounded update directions, which implies Lipschitz continuous objective for typical choices of update direction.

8.	Can you please report the language modeling results on the same figure?

9.	Could you pleas explicitly define the exact algorithm studied for Theorem 4.2 and 4.3? With several step size and update rule definitions, it is crucial to have a clear definition of the algorithm and parameters for respective theorems.

**Ethical Concerns:**

["NO or VERY MINOR ethics concerns only"]

**Final Justification:**

I am still not convinced with the aiming condition; it imposes alignment between the algorithm-specific descent direction $d_t$ (imagine when $d_t$ is a momentum estimate) and the solution of the problem. I think a more convincing study should consider aiming conditions with respect to stochastic gradients, and then use this relation to prove bounds on the alignment for $d_t$. Also, for the case when weight decay is zero we have a quadratic lower bound for the inner product in aiming condition.

The scale of the experiments is enough for a theory-oriented paper. The fact that they have both language and vision models in their experiments is important; most new designs work for only one type of models and it is hard to find a good set of parameters that makes it work in both cases.

The authors display a good understanding of the relevant literature and try to analyze a more general setup than what the literature focuses on. One might say the typical smoothness assumption is too restrictive and we need to look for relaxations, and the aiming condition could be a starting point.

**Limitations:**

There is not a dedicated section/paragraph for technical limitations of the proposed method and their results. For instance, a detailed discussion of where the Assumption 1 and 2 stand w.r.t. standard assumptions and how the Theorem 4.2 and 4.3 statements/results will change when $\lambda = 0$ are necessary additions to the paper.

**Paper Formatting Concerns:**

I have seen no issues.

**Quality:**

2

**Strengths And Weaknesses:**

I have read the manuscript and the appendices completely before writing my comments.

**STRENGTHS:**

1.	The proposed family of algorithms are derived based on the convergence analysis. The analysis is done with respect to the shrinkage of the distance to the optimum, which typical in the study of strongly convex functions (and sometimes PL functions). Although the intuition behind the heuristic simplifications are not explained well, they lead to variants which are storage-efficient (and tuning-efficient) compared to Adam.

2.	The authors dedicate a section to the explanation of their algorithm and its properties, which is important to have for new algorithm proposals.

3.	The paper does not explicitly make convexity or smoothness assumptions. (instead, there are two assumptions which are not typical in optimization and ML literature.)

4.	 There are both language modeling and image classification experiments in the paper, where Adam and proposed BCOS algorithms are contrasted.

**WEAKNESSES:**

1.	Focusing on the better performing variants; basically, the proposed algorithms simply explore different parameter updates for the second moment estimate, $v_t$. However, the intuition for how each variant is obtained seems arbitrary. For instance, for stochastic gradients $g_t$, the authors propose $E[g_t] \approx m_t$ and $E[g_t^2] \approx g_t^2$ to obtain an implementable version of the conceptual algorithm, however, this choices are not motivated. A better option would be the other way around, $E[g_t] \approx g_t$ and $E[g_t^2] \approx m_t^2$ if the goal was minimizing variance of $E[g_t^2]$ and minimizing the bias of $E[g_t]$. There is no intuition why the proposed simplification is favorable than any other.

2.	The convergence for the implementable algorithm is given in terms of asymptotic convergence of $\| x_t – x^* \|$ and the iterates converge to a *neighborhood* of the solution. This result is rather weak given that the analyzed algorithm is a variant of AdamW. For instance, a high bias bound for Assumption 2. Would yield a large diameter for the convergence with the weight decay version. For $\lambda = 0$, I am not sure how would the condition of the Theorem 4.3 is satisfied, discussion of which is missing in the paper.

3.	Regarding the previous bullet point, the analysis does not require convexity of smoothness (which is interesting), but there are two assumptions which are not typical in the literature and the discussion or verification on Assumptions 1 and 2 are not satisfactory. The aiming condition dictates alignment between the conceptual, scaled update direction and the vector $x_t - x^*$, which does not imply convex functions directly. However, when update direction is $m_t$ and the weight decay is non-zero, it is hard to interpret how strong the assumption is, as the quadratic term on the right-hand side is reminiscent of strong convexity. For $d_t = g_t$ and $\lambda = 0$, authors provide some intuition but it is not satisfactory as it is not evident which objective function classes it covers in general. If the BCOS algorithms are analyzed under a more standard assumption set (e.g., convexity, smoothness and bounded variance), it would be a fair comparison of the proposed methods against Adam.

4.	Similarly, the bias condition needs examples for verification. For instance, the two main choices, $d_t = g_t$ and $d_t = m_t$ could be explored and the bias bound could be verified with exact choices for $\tau, \epsilon$.

5.	The experiments for ViTs imply a possible test performance improvement, but it is not enough to make a concrete conclusion. The language experiments are not compared on the same figure, making it hard to assess the relative performances. I find the experiments to be inconclusive as the results are not necessarily in favor of BCOS family.

---

> ### Author Rebuttal · Authors · 2025-07-31
>
> We first address a few common questions raised by the reviewers, then respond to the specific questions by this reviewer.
>
> # Addressing several common questions
> ## 1. Aiming condition
> Aiming condition appeared in the original Robbins-Monro paper [38] on stochastic approximation as the main assumption to guarantee convergence (single dimension case). The vector version $\langle x-x_*,\mathbf{E}[g(x,\xi)]\rangle>0$ appeared in the multi-dimensional extension by Blum [3]. Both classical works require slightly stronger aiming conditions to prove almost sure convergence, and a typical sufficient condition is
> $$L\|x-x_*\|^2 \geq \langle x-x_*, \mathbf{E}[g(x,\xi)]\rangle \geq \mu\|x-x_*\|^2$$
> for some $L\geq\mu>0$. Indeed, most results in our current understanding of stochastic gradient methods can be derived as special cases of the stochastic approximation literature developed in 1950's. If $g(x,\xi)$ is the stochastic gradient of some loss function, the above aiming condition is much more general than (strong) convexity and smoothness.
>
> **Example.**
> For simplicity, consider the 1-dimensional case $x\in\mathbb{R}$ and without loss of generality let $x_*=0$. Let $g(x):=\mathbf{E}[g(x,\xi)]$ be any continuous function that satisfies $g(x) > 0$ for $x> 0$ and $g(x)<0$ for $x<0$, which leads to
> $$
> \langle x-x_*,g(x)\rangle =x \cdot g(x)>0, \qquad \forall x\neq 0.
> $$
> Suppose $g(x)$ is the gradient of some loss function $f(x)$. Then this condition is much weaker than convexity, which would require monotonicity of $g(x)$, that is, $(x-y)(g(x)-g(y))>0$ for all $x,y\in\mathbb{R}$.
> For example, consider
> $$g(x)=x(x-3)^2(x+2)^2+\lambda x,$$
> which is a non-monotone polynomial, thus it is the gradient of a nonconvex polynomial. But this function satisfies the aiming condition for $x_*=0$ and for any $\lambda\geq 0$. Apparently, such examples abound.
>
> **Our aiming condition (Assumption A).**
> When the search direction $d_t$ is independent of the past trajectory (e.g., stochastic gradient with i.i.d. noise), our aiming condition in Equation (23) is equivalent to
> $$
> \left\langle x-x_*, \frac{\mathbf{E}[d(x,\xi)]}{\sqrt{\mathbf{E}[d(x,\xi)^2]}} + \lambda x \right\rangle \geq \lambda\|x-x_*\|^2,
> $$
> where we used the definition of $\rho_t$ to expand
> $\sqrt{\rho_t}\odot\textup{sign}(\mathbf{E}_t[d_t])=\frac{\mathbf{E}_t[d_t]}{\sqrt{\mathbf{E}_t[d^2]}}=\frac{\mathbf{E}[d(x,\xi)]}{\sqrt{\mathbf{E}[d(x,\xi)^2]}}$.
>    - Our aiming condition is an extension of the classical one by incorporating coordinate-wise scaling, which is a natural adaptation to handle modern optimizers with diagonal scaling.
>    - We do not need upper bound on the inner product as in the classical aiming condition (which is implied by smoothness), due to the coordinate-wise normalization by $\sqrt{\mathbf{E}_t[d_t^2]}$.
>    - Diagonal scaling in the aiming condition causes subtle changes on the problem structure. For example, the corresponding optimal point $x_*$ is now different. This is similar to the fact that adding the regularization $\lambda\|x\|^2$ to the loss function changes the optimal point of the overall loss. Moverover,
>        * It no longer contains all convex functions as special cases, as shown by our counter example in Appendix B. We use this very ill-conditioned quadratic function to illustrate the point. The new aiming condition still holds for most convex functions that are not severely ill-conditioning (depending on the scaling). Meanwhile, diagonal scaling also opens up additional non-convex functions to satisfy the aiming condition. Therefore, it is overlapping with convexity.
>        * Since the optimal solution $x_*$ is changed with diagonal scaling, our example of ill-conditioned quadratic may still satisfy the aiming condition with respect to a different $x_*$.
>        * The question of finding the optimal solution $x_*$ given scaled directions $d(x)$ falls into the standard form of *variational inequality* $\langle x-x_*,d(x)\rangle\geq 0$ or the strengthened version $\langle x-x_*, d(x)+\lambda x\rangle\geq\lambda\|x\|^2$. There is a vast literature on solving variational inequalities and its connection to nonlinear programming. BCOS is a class of effective methods for solving *stochastic variational inequalities*.
>    - To handle the general case of correlated search directions, such as using stochastic momentum, we strengthen the aiming condition to be trajectory-dependent, as stated in Assumption A. Admittedly, such a condition is difficult to check a priori. But we have computational evidence that they mostly hold in our experiments (by recording the final weights of a trained model as $x_*$ and repeat the experiments with same random seeds in initialization and data sampling.)
>
> ## 2. Convergence analysis when $\lambda=0$
> We gave some brief comments on convergence analysis when $\lambda=0$ at the bottom of page 7. To further elaborate:
>   - When $\lambda=0$, we need slightly stronger aiming condition to provide similar convergence guarantees. For example, it is sufficient to have
>     $$
>     \left\langle x-x_*, \frac{\mathbf{E}[d(x,\xi)]}{\sqrt{\mathbf{E}[d(x,\xi)^2]}} \right\rangle \geq \mu \|x-x_*\|^2,
>     $$
>     for some $\mu>0$. This can be the case of assimilating $\lambda x$ into the search direction $d(x,\xi)$, where we do not get the full strength of regularization $\lambda$ on the right hand side (due to diagonal scaling), but only settle for some $\mu<\lambda$.
>   - For convergence rate, with any $\mu>0$ in the above condition, we can still obtain $O(1/t)$ rate for the conceptual algorithm. Moverover, we can remove the condition $\alpha\lambda>1/2$ and settle with $O(1/t^p)$ where $p<1$ can be arbitrarily close to 1. These are all direct consequences of classical stochastic approximation [6].
>
> ## 3. Bias and variance bounds for the second-moment estimator
> Our analysis of the practical BCOS algorithms relies on Assumption B. We can always construct an estimator with $\tau<1$ with sufficiently large $\epsilon$.
>    - We listed the bias-variance trade-offs of several popular algorithms at the bottom of page 8. Apparently, more concrete bounds can be derived depending on $\tau$ and $\epsilon$, as well as the smoothness constants of the loss function. But their expressions can be quite tedious and loose. We will try to include some refined bounds in the revision, but they do not dictate our fundamental contributions.
>    - The exact magnitudes of the bias and variance of the estimators do not affect stability of the algorithms. The stability or convergence of the algorithms are determined by the aiming condition for the conceptual algorithm. When the conceptual algorithm is stable and converge, the actual bias and variance of the practical algorithms only affect the quality of solution (radius of neighborhood of $x_*$).
>
>
> # Answering specific questions of the reviewer
>
> 1. In deriving the conditional estimator of BCOS (Section 3.2), we want to have small bias and variance for estimating $\mathbf{E}_t [m_t^2]$. Giving its expansion in Equation (19), a low-variance estimator for $\mathbf{E}_t[g_t]$ is apparently $m_t$ (although biased). We also want a low-variance estimator for $\mathbf{E}_t[g_t^2]$, but it requires another EMA estimator. So we settle with the unbiased estimator $g_t^2$ itself, which may have higher variance, but it is attenuated by the factor $(1-\beta)^2$. These are the reasoning behind our choices.
>
> 2. Convergence rate in Theorem 4.2 has dimension dependence. This is caused by using coordinate-wise analysis. Such dimension dependence actually exists for most convergence rates for coordinate descent methods. Here we do update all coordinates simultaneously but do not know how to remove this dependence.
>
> 3. Optimize stepsize before taking expectations will lead to very volative stepsizes. It does not make sense trying to minimize the distance for every instantaneous iterate; what matters is the expected distance.
>
> 4. On the second line of the equation below line 602, there is a typo: a minus sign is missing in the second absolute value, it should be the absolute value of the diﬀerence of the two terms, not multiplication.
>
> 5. We explained the analysis for the case $\lambda=0$ in addressing common questions. No change needed for the variance and bias assumption (Assumption B).
>
> 6. Example given in answers to common questions. Let $g(x) = x(x−3)^2(x+2)^2 + \lambda x$, then $f(x)$ is obtained by integration of $g(x)$. It is clear that such examples are abundant. For $d_t=m_t$, we do not have a clean analytic example, and need to resort to empirical verification.
>
> 7. Assuming bounded variance on $g_t$ will certainly lead to bounded variance on $m_t$. But we need to assume bounded variance on $g_t^2$. Indeed, with the assumption that $g_t^2$ or $d_t^2$ has bounded variance, we can remove the assumption that $d_t$ in bounded almost surely in Theorem 4.3. We will incorporate this in the revision.
>
> 8. Figure 2 shows results for three variants of BCOSW and AdamW on the same plot.
>
> 9. Theorem 4.2 clearly states that it is about the conceptual method in Equation (22). Theorem 4.3 is about the practical method in line 219-220. We will add the explicit reference.
>
> # Main contributions
>
> Finally, we remind the review of our main contributions:
> 1. Derive BCOS family from simple principle of coordinate-wise contraction, which gives Adam(W) a novel interpretation.
> 2. We develop a new variant by leveraging a simple conditional estimator, which obtains competitive performance against AdamW but only use half of its optimizer states and less hyper parameter to tune.
> 3. We provide convergence of a broad family of algorithms using an *Aiming Condition*, which reintroduce the rich literature on stochastic approximation developed from its birth in the 1950's to the contemporary research community.
> 4. We develop a new, two-stage analysis technique (from conceptual to practical) for analyzing general stochastic approximation algorithms.

---

> > ### Comment · Reviewer_SALE · 2025-08-02
> > **Response to rebuttal**
> >
> > Thank you for the detailed explanations on the aiming condition, and your rigorous response on the theory related questions.
> >
> > **Aiming condition:** Let me summarize my concerns on it. (1) it considers the expected direction defined by $d_t$, an algorithm dependent quantity, rather than the stochastic gradients themselves, which would be oracle-related and more reasonable (2) The condition implies a quadratic lower bound when $\lambda = 0$; the condition is tied to weight decay and in my opinion, it more reasonable to have conditions independent of the implementation-specific dynamics.
> >
> > As the authors point out, the condition does not subsume all convex functions but a subset of the class. I understand the authors try to focus on an exploratory problem with the aiming condition, which is important for opening up new horizons. However, it is not clear how general it is and where it stands in the literature with respect to standard problem setups. The theorem statement for the implementable algorithm considers asymptotic convergence to the neighborhood of the solution.
> >
> > It seems that the condition implies there exists only a single global minimizer of the function if I am not mistaken. Is this true?
> >
> > **2:** Let me be more specific with the dimension dependence. My expectation would be $\sqrt{n}$ dependence, similar to what preconditioned methods get for per-coordinate scaling. I am still curios what values to you pick exactly for $\alpha$ and $\lambda$ to balance the dimension dependence.
> >
> > **7:** Do you have a result on this or do you propose this as a future work? I would be interested in seeing any results you have on it at the moment. Otherwise, it would be too much to ask at this point of discussion period if you do not have a complete proof already.
> >
> > **9:** I understand that 4.2 is for the conceptual algorithm and 4.3 for the implementable version. I was asking about the exact update rules and also the choice of algorithm parameters that guarantees the presented bounds.

---

> > > ### Author Response · Authors · 2025-08-05
> > > **Aiming condition and "standard" problem setup**
> > >
> > > Thank you for your feedback for our rebuttal. Below are our further clarifications.
> > > # On Aiming Condition
> > >   (1) The aiming condition (Assumption A) is general. It covers the special case of $d_t$ being stochastic gradient, i.e., $d_t=\nabla f(x_t,\xi_t)$. There, Assumption A is only oracle-based and not trajectory-dependent, so we can remove subscripts $t$ in Assumption A:
> > >   $$
> > >   \left\langle x-x_*, \frac{\mathbf{E}[\nabla f(x,\xi)]}{\sqrt{\mathbf{E}[\nabla f(x,\xi)^2]}} + \lambda x \right\rangle \geq \lambda\|x-x_*\|^2,\qquad \forall x\in\mathbb{R}^n.
> > >   $$
> > >   This simplified condition is presented in our rebuttal and commented on in Line 201. The specific form of this condition makes it more convenient to directly compare it with (strong) convexity. We will separate this special case of in the paper revision to improve clarity.
> > >
> > >   We employ a more general, trajectory-dependent aiming condition in Assumption A to include trajectory-dependent $d_t$, e.g. momentum. Unfortunately, we do not have a clean oracle-based analytic condition, but will add plots that illustrate the aiming condition in the numerical examples.
> > >
> > >   (2) When $\lambda=0$, a weaker condition guarantees almost-sure convergence. Specifically, when $d_t=\nabla f(x_t,\xi_t)$,
> > >   $$
> > >   \left\langle x-x_*, \frac{\mathbf{E}[\nabla f(x,\xi)]}{\sqrt{\mathbf{E}[\nabla f(x,\xi)^2]}} \right\rangle \geq \phi(\|x-x_*\|),\qquad \forall x\in\mathbb{R}^n,
> > >   $$
> > >   for any continuous and strictly positive function $\phi$.
> > >
> > >   When the weight decay ($L_2$ regularization) is not decoupled but absorbed into $d_t$, then we land on the case in rebuttal, Common Questions 2. Such discussions will be added to the revision.
> > > # On generality of aiming condition and "standard" problem setup
> > > Due to the generality of the aiming condition, we believe it is more foundamental than the standard setup of convexity and smoothness. We consider leading such a change of paradigm (back to the 50's) as one of our major contributions.
> > >    -  With a common scalar stepsize for all coordinates, $s_{t,k}=\gamma_t$, the aiming condition becomes strictly more general than (strong) convexity, and it is the foundation for all the classical stochastic approximation literature, starting from Robbins and Monro [38]. Most of our current understanding of stochastic gradient methods can be derived as special cases of these early developments.
> > >    - Assumption A extends the classical condition [38] by incorporating (block) coordinate-wise stepsizes, $s_{t,k}=\gamma_{t,k}1_{n_k}$. Due to this change, convexity no longer implies aiming. Nevertheless, as shown in Appendix B, the exception only occurs for convex functions with severe cross-coordinate ill-conditioning (because diagonal scaling has limited capability in coping with cross-coordinate ill-conditioning). Luckily, this does NOT limit the applicability of our assumption: the corresponding variational inequality may still has a well-defined solution $x_*$.
> > >    - An improvement over the classical aiming condition is the dismissal of a quadratic upper bound on the inner product (see rebuttal, common question 1), due to the proper scaling of $d_t$ by $\sqrt{\mathbf{E_t}[d_t^2]}$. This is roughly equivalent to removing the smoothness condition, representing another major deviation of our theory from "standard" setting of convexity and smoothness.
> > >   - We hope this paper can raise awareness of the rich literature on stochastic approximation developed in the 50's, which contain more general and powerful tools that are mostly ignored. They provide an alternative, if not a more foundamental understanding of state-of-the-art optimizers such as Adam/AdamW, and lead to novel variants such as BCOS with conditional estimators.
> > > # Clarification of specific questions.
> > >   0 The aiming condition implies a single global equilibrium point $x_*$ (not necessarily the global minimizer for a nonconvex function) for the conceptual algorithms (using expected 2nd moment). But with the practical algorithms, $x_t$'s only converge to a neighborhood of the equilibrium point, and the size of the neighborhood depends on the quality of the 2nd moment estimator.
> > >
> > >   2 We only manage to control $c_*$ to be order $n$, as we do not use preconditioning type of arguments. The condition $\alpha\lambda>1/2$ does limit capability of control the constant in $O(1/t)$. On the other hand, we can remove this condition and use arbitrary $\alpha$ that satisfies $\alpha\lambda<1$, then have to settle with an asymptotic rate of $O(1/t^p)$ for $0<p<1$. We can include such results in the revision.
> > >
> > >   7 We have a concrete proof. Specifically, in the proof of Theorem 4.3, we only need the inequality above Line 635 to hold, which is sufficient by assuming bounded variance on $d_t$.
> > >
> > >   9 For Theorem 4.3, the learning rate $\{\alpha_t\}$ need to satisfy Equation (25), $\gamma_t$ is given by Equation (18), $d_t$ can be any direction that satisfies Assumption A, and $v_t$ in (18) needs to satisfy Assumption B.

---

> > > > ### Comment · Reviewer_SALE · 2025-08-06
> > > > **Response to authors**
> > > >
> > > > Thank you for the detailed, clean and explanatory response.
> > > >
> > > > Focusing on the proposed aiming condition and the coordinate-wise, pre-conditioned algorithm template, there are advantages and generalizations compared to the classical stochastic approximation literature. The authors identify some of them during the discussion period. The proposed setting is beyond the Lipschitz smooth class of functions, more general than typical assumption that prevails in the optimization literature.
> > > >
> > > > My concern related to the aiming condition still stands regarding the use of possibly momentum-based update direction $d_t$ and the case when $\lambda = 0$. Moreover, the practical algorithm guarantees convergence to a neighborhood of the solution rather than to a solution.
> > > >
> > > > To understand your point about convex problems and ill-conditioining, let's imagine a quadratic function  of the form $f(x) = x^T A x$. What choice of $A$ would be problematic?
> > > >
> > > > I must say my overall thoughts are more positive than my initial evaluation, but I want to discuss my concerns with the other reviewers, who gave better scores than mine, and the AC before making a conclusive decision.

---

> > > > > ### Author Response · Authors · 2025-08-07
> > > > >
> > > > > We thank the reviewer for the effort in understanding our contributions and rebuttal, and giving a more positive evaluation of our work. Here are our clarifications regarding the reviewer's concerns.
> > > > >
> > > > > - *"My concern related to the aiming condition still stands regarding the use of possibly momentum-based update direction $d_t$ and the case when $\lambda=0$."*
> > > > >
> > > > > As we explain in the rebuttal, the aiming condition in Assumption A is quite general. We will single out and emphasize the case of $d_t$ being stochastic gradient, which has a clean, trajectory independent form that can be directly compared with convexity. When $d_t$ is the momentum, we need the trajectory dependent form, which does not have a clean analytic exposition, but we do observe it holds in our experiments and will add empirical evidence.
> > > > >
> > > > > The convergence analysis of $\lambda=0$ is more subtle as we explained in rebuttal and further clarifications above. We do have a full set of convergence analysis results for this case, but chose not to include in the paper due to limited space allowed. Also, in practice, we almost always use $\lambda>0$, so we focused on the analysis that is most relevant and easier to present. The $\lambda=0$ case is more subtle but the fundamental ideas and tools are the same.
> > > > >
> > > > > - *"Moreover, the practical algorithm guarantees convergence to a neighborhood of the solution rather than to a solution."*
> > > > >
> > > > > Our analysis of the practical algorithms guarantees convergence to a neighborhood of the solution rather than the solution itself. But this is about Almost-Sure convergence, much stronger than convergence in expectation! Moreover, we derived the dependence of the size of the neighborhood on both the bias and variance of the 2nd-moment estimator. In particular, when both bias and variance vanish, our analysis shows the size of the neighborhood shrinks to zero (LHS of Equation (29) is equal to zero as the bias and variance vanish, leading to a zero-radius neighborhood).
> > > > >
> > > > >
> > > > > **This type of convergence guarantee is a natural outcome of our new framework for convergence analysis of general stochastic approximation algorithms, which we call "Statistical Stochastic Approximation" (SSA). The classical framework of stochastic approximation or stochastic optimization, after taking stochastic gradients from an oracle, focuses on how to construct specific update directions (such as momentum) and how to choose the (coordinate-wise) stepsizes. The discovery of new algorithms is often based on ad hoc heuristics and the analysis needs to be tailored to each specific variants. Our SSA framework is based on simple and clean principle (block coordinate-wise contraction) and formally introduces statistical estimators for key entities such as the second-moment of the search direction. This framework gives a novel, unified interpretation of many effective algorithms and clear guidance for future algorithm development. Our two-stage analysis framework (from conceptual to practical algorithms) applies to a broad family of algorithms and gives explicit characterization of the effect of bias-variance trade-offs of different estimators.**
> > > > >
> > > > > **On top of the above points, our conditional BCOS-c method demonstrates competitive performance against AdamW but only use half of its optimizer states and fewer hyper-parameters to tune. This alone makes a novel and strong contribution. It is one of the very few essential progresses on optimizers since Adam and AdamW.**
> > > > >
> > > > > These contributions are of fundamental nature and far from being incremental, so it may take time for acceptance by the research community. We believe NeurIPS should be the perfect venue to promote new ideas and methodologies, and hope the reviewer would agree that our work passes the bar.
> > > > >
> > > > > - *"To understand your point about convex problems and ill-conditioining, let's imagine a quadratic function of the form $f(x)=x^T A x$. What choice of $A$ would be problematic?"*
> > > > >
> > > > > If $A$ is diagonal and positive definite (positive diagonals), then Aiming condition holds with arbitrary diagonal scaling, even with severe ill-conditioning (orders of magnitude difference of diagonal entries). The aiming condition may not hold with respect to $x^*=0$ if A is ill-conditioned with strong cross-coordinate coupling, as shown by our example at the end of Appendix B. Notice that this example is with specific coordinate-wise scaling that corresponds to sign-gradient method. In the high-dimensional setting or over-parametrized regime, such cases may be rare (especially with milder scaling than the sign() function). This is only our intuition, and there are certainly deeper theoretical investigations required for future work.

---

### Comment · Area_Chair_EDPw · 2025-08-05

Dear Reviewers,

Thank you again for your time and efforts in reviewing papers for NeurIPS 2025.

I am writing to remind you that **active participation in the author-reviewer discussion phase is mandatory**. According to the guidelines from the NeurIPS program chairs, reviewers are **required to engage directly with the authors in the discussion thread**, especially in response to their rebuttals.

Please note the following important policy:

- Simply reading the rebuttal or internally considering it is **not sufficient** -- reviewers must **post at least one message to the authors**, even if it is only to confirm that their concerns were resolved. If they have not been addressed, please explain why.

- **Acknowledging the rebuttal without any engagement with the authors will be considered insufficient**. I am obligated to flag such cases using the *InsufficientReview* mechanism, which may **impact future reviewing invitations and result in desk rejection of your own submissions**.

If you have not yet responded to the authors in the discussion thread, I kindly ask you to do so **as soon as possible**, and **no later than August 8, 11:59pm AoE**.

Please don't hesitate to reach out to me if you have any questions or concerns.

Best regards,

AC

---

### Note · Authors · 2025-08-15

We thank the reviewers for their feedback and discussions and the area chair for coordinating the entire review process. We hope the reviewers would agree that we have clarified the main technical questions (on the aiming condition, almost sure convergence to a neighborhood, and analysis for $\lambda=0$, etc.).
Certainly, the algorithms and theory we present in one paper are far from complete in addressing many open questions in stochastic optimization of deep learning models.
However, we would like to emphasize that our contributions are of fundamental nature and far from incremental.

  1. The BCOS framework we introduce is based on the simple principle of block-coordinate contraction, which provides novel interpretations and a unified analysis of many popular and most effective stochastic optimization algorithms. Moreover, this framework leads to a new class of algorithms based on conditional estimators of the 2nd moment; in particular, the BCOSW-c variant achieves competitive performance against AdamW but requires half of the optimizer states and fewer hyper-parameters. This is one of the very few fundamental progresses on optimizers since Adam and AdamW.
   2. As important as convexity and smoothness in optimization research, they are fundamentally limited in addressing convergence analysis of adaptive optimizers that dominate modern deep learning. This paper breaks away from this limitation by introducing a novel aiming condition and formal statistical tools in the design and analysis of stochastic optimization methods. Along the way, we uncover the powerful theory developed in the classical stochastic approximation literature. We believe this will open up new avenues of research and further accelerate progress in this very exciting area.

---

### Decision · Program_Chairs · 2025-09-17

**Decision:**

Reject

**Comment:**

**Summary.** This paper introduces a family of stochastic optimization methods with block-coordinate stepsizes. The ideal stepsizes are designed to minimize the distance from the next iterate to the optimal point. Since these stepsizes cannot be computed in practice, the authors propose implementable variants that estimate the second moment of the stochastic gradient using exponential moving averages. Under the so-called aiming condition, the authors establish an $\mathcal{O}(1/t)$ convergence rate for the conceptual (non-implementable) method with weight decay and prove convergence to a neighborhood for the practical version. The paper also includes extensive numerical experiments, which illustrate the advantages of the proposed methods over standard baselines.


**Strengths and weaknesses.** The proposed algorithm is well motivated and supported by clear theoretical derivations. The numerical results are comprehensive for a theory-focused paper and demonstrate the empirical benefits of the approach.

On the other hand, the main theoretical guarantee for the practical variant establishes only convergence to a neighborhood rather than to the optimum. In addition, the aiming condition employed in the analysis is algorithm-specific, which makes it difficult to compare the theoretical guarantees directly with existing results.

**Reviewers' consensus.** Reviewers 6Z8J and SALE gave borderline rejection scores (3), emphasizing the aforementioned weaknesses. Reviewers X2fe and XLQ3 gave borderline acceptance scores (4), highlighting the aforementioned strengths of the work. Overall, the reviews reflect a lack of clear consensus, with opinions split along the identified strengths and weaknesses.

**Final recommendation.** After carefully considering the reviews, rebuttal, and my own assessment, I find the weaknesses significant enough to warrant a major revision. Therefore, I recommend rejection at this stage. Nevertheless, the paper presents an interesting and promising idea, and I encourage the authors to address the identified limitations and resubmit to a future venue.